# Towards User-level Private Reinforcement Learning with Human Feedback

**Jiaming Zhang**[1,2,3]* **Mingxi Lei**[4]* **Meng Ding**[2,4]**, Mengdi Li**[2,3]**,Zihang Xiang**[2,3]**,
Difei Xu**[2,3]**, Jinhui Xu**[4]**, Di Wang**[2,3]
[1]Renmin University of China, [2]Provable Responsible AI and Data Analytics (PRADA) Lab,
[3]King Abdullah University of Science and Technology, [4]State University of New York at Buffalo
**Correspondence:** di.wang@kaust.edu.sa

## Abstract

Reinforcement Learning with Human Feedback (RLHF) has emerged as an influential technique, enabling the alignment of large language models (LLMs) with human preferences. However, how to protect user preference privacy has become a crucial issue, as LLMs tend to remember users' preferences. Most previous work has focused on using differential privacy (DP) to protect the privacy of individual data. However, they have concentrated primarily on item-level privacy protection and have unsatisfactory performance for user-level privacy, which is more common in RLHF. This study proposes a novel framework, AUP-RLHF, which integrates user-level label DP into RLHF. We first show that the classical random response algorithm, which achieves an acceptable performance in item-level privacy, leads to suboptimal utility when in the user-level settings. We then establish a lower bound for the user-level label DP-RLHF and develop the AUP-RLHF algorithm, which guarantees $(\varepsilon, \delta)$ user-level privacy and achieves an improved estimation error. Experimental results show that AUP-RLHF outperforms existing baseline methods in sentiment generation and summarization tasks, achieving a better privacy-utility trade-off.

## 1 Introduction

The advent of large language models (LLMs) has significantly transformed the field of artificial intelligence, leading to widespread adoption and application in diverse domains(Hong et al., 2024; Yang et al.; Cheng et al.; Zhang et al., 2024), such as legal (Chalkidis & Kementchedjhieva, 2023; Wu et al., 2023b), medical (Thirunavukarasu et al., 2023; García-Ferrero et al., 2024; Yuan et al., 2024), and coding assistant (Ross et al., 2023; Nam et al., 2024; Kazemitabaar et al., 2024). A notable advancement in this area is the incorporation of Reinforcement Learning from Human Feedback (RLHF), a paradigm that improves the performance and alignment of language models by integrating human preferences and feedback into the training process (Ouyang et al., 2022).

However, since RLHF relies on collecting human feedback, it inevitably raises privacy concerns, particularly regarding the identity and sensitive information of human annotators. For example, in the financial domain, investment advisors can annotate and provide feedback on LLM-generated investment recommendations. These preference labels may encompass sensitive information such as individual investment strategies, risk preferences, and financial objectives. In addition, LLMs have the potential to infer individual user preferences, interests, or personal attributes by analyzing queries and interactions posed. Such inferences may lead to the generation of targeted advertisements, personalized suggestions, or other customized content, which in turn could inadvertently expose sensitive aspects of the user's private life (Yan et al., 2024; Lyu et al., 2023; Harte et al., 2023; Hu et al., 2024; Huang et al., 2024).

---

*Equal contributions.

To address the aforementioned privacy concerns, Differential Privacy (DP) (Dwork et al., 2006), which quantifies privacy protection by adding noise to ensure the anonymity of individual data in statistical analysis, has become a common approach, effectively protecting user privacy. Recently, there have been an increasing number of studies focusing on DP-RLHF (Chowdhury et al., 2024b; Chua et al., 2024; Wu et al., 2023a; Teku et al., 2025), demonstrating the feasibility of DP to RLHF. However, traditional DP algorithms in RLHF focus on item-level privacy, where they ensure that the inclusion or exclusion of any single data point does not significantly affect the model's output. In practical scenarios, a user typically contributes multiple pieces of data, rendering item-level privacy insufficient. Specifically, existing methods offer privacy guarantees that deteriorate as user participation increases or blindly introduce excessive noise, leveraging the group property of differential privacy, which severely impacts the performance of the model in deployment (Levy et al., 2021). For example, in Section 4, we will show that the Randomized Response mechanism, which is the canonical way of protecting label privacy and achieves the nearly optimal rate in the item-level case (Chowdhury et al., 2024b), will significantly degrade its utility in the user-level privacy scenario. Thus, there is a pressing need for developing user-level label DP-RLHF with a better privacy-utility tradeoff.

To tackle the challenges outlined above, motivated by the current developments of user-level DP supervised learning (Geyer et al., 2017; Bassily & Sun, 2023a; Levy et al., 2021; Liu & Asi, 2024a; Ghazi et al., 2024; Zhao et al., 2024), we propose a novel framework, namely AUP-RLHF, which satisfies user-level label DP and achieves a smaller estimation error. The key intuition is the use of the average gradient of the loss with respect to each user's preferences for parameter updates in the DP-SGD algorithm (Abadi et al., 2016; Su et al., 2024; Shen et al., 2023; Xiao et al., 2023; Hu et al., 2022). However, unlike the existing DP-SGD designed for item level, we leverage outlier removal and adaptive sampling processes to handle the high sensitivity to ensure that the gradient exhibits concentrated properties. Then, at each step of the SGD update, we add noise, which is significantly small, related to the concentrated parameters only to ensure privacy. Our contributions are as follows:

1. We first demonstrate that under canonical assumptions, the classical randomized response method, which performs well in the item-level setting, exhibits a large estimation error $O(\frac{d\sqrt{m}}{\sqrt{n}\epsilon})$ in the user-level setting. Specifically, when the contribution $m$ for each user is large, randomized response sacrifices utility to provide uniform privacy protection in the privacy-utility trade-off, rendering the algorithm unsuitable. We also establish a lower bound $\Omega\left(\frac{d}{\sqrt{nm}} + \frac{\sqrt{d}}{\sqrt{mn}\epsilon}\right)$ for user-level DP-RLHF.

2. We are the first to conduct a theoretical analysis of user-level differential privacy in RLHF. To close the gap between the upper and lower bounds, we develop AUP-RLHF that satisfies $(\varepsilon, \delta)$-user-level label privacy, with an upper bound of $O\left(\frac{d\sqrt{d}}{\sqrt{mn}\varepsilon}\right)$ for the estimation error. Additionally, we extend the algorithm to the $K$-wise case, where the upper bound of the estimation error remains $O\left(\frac{K^2 d\sqrt{d}}{\sqrt{mn}\varepsilon}\right)$.

3. We empirically validate AUP-RLHF through experiments on commonly studied tasks. We conducted controlled sentiment generation and summarization experiments. We show that, across base models of varying sizes and different privacy parameter configurations, AUP-RLHF consistently outperforms the other user-level DP baselines.

## 2 Related Work

**User-level DP learning.** User-level DP has garnered increasing attention due to its privacy protection, which aligns more closely with real-world scenarios. Several studies have explored various tasks of user-level DP, such as mean estimation (Levy et al., 2021), empirical risk minimization (Levy et al., 2021), and stochastic convex optimization (Bassily & Sun, 2023b; Liu & Asi, 2024b). Levy et al. (2021) first introduced user-level DP, providing stricter privacy protection by safeguarding the entire contribution of users, based on a novel private mean estimation algorithm. Bassily & Sun (2023b) further investigated how

carefully selected first-order optimization methods can achieve optimal utility under local DP conditions. Liu & Asi (2024b), on the other hand, ensured query concentration through above-threshold and adaptive sampling techniques, thereby releasing the minimal number of user assumptions while still achieving optimal utility. However, none of them has considered RLHF. In this paper, we build the first theoretical results on user-level (label) DP-RLHF.

**Privacy-preserving in RLHF.** Zhu et al. (2023) provided a theoretical framework for RLHF under clean data conditions, demonstrating the properties of MLE and pessimistic MLE. However, due to privacy leakage issues (Li et al., 2023), MLE cannot be applied directly. Building on this, Chowdhury et al. (2024b) introduced a theoretical framework for incorporating DP into RLHF, designing an unbiased loss function based on random responses to achieve DP. Teku et al. (2025) proposed a novel multi-stage mechanism that privately aligns the model with labels from previous stages combined with random responses. However, as we will show later, random responses cannot be easily extended to more practical user-level settings, which results in poor utility. Meanwhile, Wu et al. (2023a) utilized DP-SGD across the three stages of RLHF—Supervised Fine-Tuning, Reward Model Learning, and Alignment with PPO—to ensure privacy preservation. Chua et al. (2024) explored the application of Group Privacy and User-wise DP-SGD to achieve user-level DP in RLHF. However, they provide only experimental results and lack rigorous theoretical guarantees. Our experiments will show that our proposal method has better performance than these methods.

## 3 Preliminaries

### 3.1 RLHF

In the RLHF process, the data structure used is a quadruple $(s, a^0, a^1, y)$, where $s$ represents the prompt provided by the user to the LLM, $a^0$ and $a^1$ are the two responses generated by the LLM, and $y \in \{0, 1\}$ indicates the human preference between the two responses. We follow the notion of Chowdhury et al. (2024b); Ouyang et al. (2022); Zhu et al. (2023), where preferences are modeled as Bradley-Terry-Luce (BTL) model:

$$\mathbb{P}_{\theta^*}\left[y = l \mid s, a^0, a^1\right] = \frac{\exp\left(r_{\theta^*}\left(s, a^l\right)\right)}{\exp\left(r_{\theta^*}\left(s, a^0\right)\right) + \exp\left(r_{\theta^*}\left(s, a^1\right)\right)} = \sigma(r_{\theta^*}(s, a^l) - r_{\theta^*}(s, a^{1-l})),$$

where, $r_{\theta^*} : \mathcal{S} \times \mathcal{A} \to \{0, 1\}$ denotes the ground-truth reward model parameterized by $\theta^*$ and $\sigma(z) = 1/(1 + e^{-z})$ refers to the sigmoid function. And we assume reward model a linear function: $r_{\theta^*}(s, a) = \phi(s, a)^\top \theta^*$, where $\phi : \mathcal{S} \times \mathcal{A} \to \mathbb{R}^d$ is some known and fixed feature map, constructed by removing the last layer of a pre-trained language model in LLM, and in that case, $\theta^*$ corresponds to the weights of the last layer. Let $x = \phi(s, a^1) - \phi(s, a^0)$ denote the differential feature of actions $a^1$ and $a^0$ at state $s$.

### 3.2 Differential Privacy

We use the notation $Z_i := (s_{i,j}, a^0_{i,j}, a^1_{i,j}, y_{i,j})^m_{j=1}$ to represent user $i$'s contributions, and $z_i := (s_{i,j}, a^0_{i,j}, a^1_{i,j}, y_{i,j})$ to represent user $i$'s $j$th contributions. We use capital Z to denote one user and z to denote one item. We consider user-level DP, which protects all of a user's contributions, meaning that the output of an algorithm $M$ operating on a dataset with $n$ users (thus, $mn$ samples in total) $D = (Z_i)^n_{i=1}$ is 'indistinguishable' when a single user's contributions in $D$ are altered. A formal definition is provided below.

**Definition 1. (User-Level Differential Privacy).** A mechanism $\mathcal{M} : (\mathcal{Z}^m)^n \to \mathbb{R}^d$ is $(\varepsilon, \delta)$ user-level differentially private, if for any neighboring datasets $\mathcal{D}, \mathcal{D}' \in (\mathcal{Z}^m)^n$ that differ in one user, and for any event $O$ in the range of $\mathcal{M}$, we have

$$\Pr[\mathcal{M}(\mathcal{D}) \in O] \leq e^\varepsilon \Pr[\mathcal{M}(\mathcal{D}') \in O] + \delta.$$

The original definition of DP (Dwork et al., 2006) assumes that the whole dataset is private, which is quite strong for many scenarios in RLHF. As in RLHF, a user's preferences are only associated with the label of the $(prompt, response, label)$ tuple, while the prompt itself is not sensitive, as it is sampled from pre-collected datasets that are already considered public knowledge (Chowdhury et al., 2024b; Teku et al., 2025). Thus, in this paper, we consider the label DP (Ghazi et al., 2021). We assume that $(s, a^0, a^1)$ is publicly accessible, while user preference data $y \in \{0, 1\}$ are private. Therefore, the focus is on protecting the user's preference information $y$. The definition of label DP at the user level is provided below.

**Definition 2. (User-Level Label Differential Privacy).** A mechanism $\mathcal{M} : (\mathcal{Z}^m)^n \to \mathbb{R}^d$ is $(\varepsilon, \delta)$ user-level label differentially private, if for any neighboring datasets $\mathcal{D}, \mathcal{D}' \in (\mathcal{Z}^m)^n$ that differ in the labels of one user, and for any event $O$ in the range of $\mathcal{M}$, we have

$$\Pr[\mathcal{M}(\mathcal{D}) \in O] \leq e^\varepsilon \Pr[\mathcal{M}(\mathcal{D}') \in O] + \delta.$$

Note that item-level label differential privacy is a specific case of this definition with $m = 1$.

## 4 Sub-optimality of Random Response

To achieve label-level DP, a natural approach is to employ the random response (RR) (Warner, 1965) mechanism to each label, which achieves privacy by randomly flipping labels. In fact, such a simple idea has been shown to achieve satisfactory performance in the item-level DP-RLHF both theoretically and practically (Teku et al., 2025; Chowdhury et al., 2024b). However, the theoretical behavior of RR in the user-level setting remains unclear. In the following, we first investigate the theoretical behavior of RR in the user-level setting. We will then show how the upper bound of the estimation error degrades when each user contributes a large amount of data $m$.

### 4.1 User-level Random Response

Let $\varepsilon \geq 0$, $m$ be the sample numbers of a single user, and $y \in \{0, 1\}$ be the true label. By the group privacy property, to extend the classical RR to the user level, we need to flip each label to make it satisfy $\frac{\varepsilon}{m}$-DP. In detail, the outputs of the RR mechanism $\tilde{y}$ follow the following probability distribution

$$\mathbb{P}[\tilde{y} = y] = \frac{e^{\varepsilon/m}}{1 + e^{\varepsilon/m}} = \sigma(\varepsilon/m), \quad \mathbb{P}[\tilde{y} \neq y] = 1 - \sigma(\varepsilon/m).$$

It is easy to show that **User-level RR** is $\varepsilon$-*user-level* label DP. Then, we consider the following loss on the perturbed data, which is unbiased to the original loss in (non-private) RLHF. We will present the design of this loss function in Appendix B.3.

$$\widehat{l}_{\mathcal{D},\varepsilon}(\theta) = - \sum_{i=1}^n \sum_{j=1}^m \left[ \mathbb{1}\left( \widetilde{y}_{i,j} = 1 \right) \log \widehat{p}_{i,j}^1 + \mathbb{1}\left( \widetilde{y}_{i,j} = 0 \right) \log \widehat{p}_{i,j}^0 \right], \tag{1}$$

where the predicted scores of each randomized label $\widetilde{y}_{i,j}$ given $x_{i,j}$ are defined as:

$$\widehat{p}_{i,j}^1 = \frac{\sigma\left(x_{i,j}^\top \theta\right)^{\sigma(\varepsilon/m)}}{\left(1 - \sigma\left(x_{i,j}^\top \theta\right)\right)^{(1-\sigma(\varepsilon/m))}}, \quad \widehat{p}_{i,j}^0 = \frac{\left(1 - \sigma\left(x_{i,j}^\top \theta\right)\right)^{\sigma(\varepsilon/m)}}{\sigma\left(x_{i,j}^\top \theta\right)^{(1-\sigma(\varepsilon/m))}}.$$

Thus, our private model is defined as

$$\widehat{\theta}_{\text{RR}} \in \operatorname{argmin}_{\theta \in \Theta_B} \widehat{l}_{\mathcal{D},\varepsilon}(\theta), \tag{2}$$

where $\widehat{\theta}_{\text{RR}}$ satisfies $\varepsilon$-*user-level* label DP due to RR and the post-processing property of DP. To get the estimation error of $\widehat{\theta}_{\text{RR}}$, we make the following assumption, which is standard in the existing literature (Shah et al., 2016; Shin et al., 2023; Chowdhury et al., 2024b).

**Assumption 1** (Boundedness). (a) $\theta^*$ lies in the set $\Theta_B = \left\{ \theta \in \mathbb{R}^d \mid \langle \mathbf{1}, \theta \rangle = 0, \|\theta\| \leq B \right\}$ with some constant $B$. Here the condition $\langle \mathbf{1}, \theta \rangle = 0$ ensures identifiability of $\theta^*$. (b) Features are bounded, i.e., for all $(s, a)$ we have $\|\phi(s, a)\| \leq L$ for some constant $L$.

Let $\Sigma_{\mathcal{D}} := \frac{1}{mn} \sum_{i=1}^{n} \sum_{j=1}^{m} x_{i,j} x_{i,j}^\top$ denote the sample covariance matrix of differential features $x_{i,j} = \phi(s_{i,j}, a_{i,j}^1) - \phi(s_{i,j}, a_{i,j}^0)$.

**Theorem 1.** *For any $\varepsilon > 0$, the private model $\hat{\theta}_{\text{RR}}$ is $\varepsilon$-DP. Moreover, under Assumption 1, for any $\alpha > 0$, with probability at least $1 - \alpha$, we have*

$$\left\| \hat{\theta}_{\text{RR}} - \theta^* \right\|_2 \leq O\left( \frac{1}{\gamma \sqrt{\lambda_{\min}(\Sigma_{\mathcal{D}})}} \frac{e^{\varepsilon/m} + 1}{e^{\varepsilon/m} - 1} \sqrt{\frac{d + \log(1/\alpha)}{nm}} \right),$$

*where $\gamma = \frac{1}{2 + e^{-2LB} + e^{2LB}}$, $\lambda_{\min}(\Sigma_{\mathcal{D}})$ is the minimum eigenvalue of $\Sigma_{\mathcal{D}}$.*

Note that, the bound is non-trivial only when the sample covariance matrix is positive definite. However, this assumption is too strong due to the high dimensionality of the feature vector. We can relax it by imposing that the population covariance matrix is positive definite by imposing a coverage assumption on the state-action feature space, which is commonly encountered in offline bandit and reinforcement learning settings (Yin et al., 2022).

**Assumption 2** (Coverage of feature space). Defining the differential state-action features $x = \phi\left(s, a^1\right) - \phi\left(s, a^0\right)$, and population covariance matrix $\Sigma = \mathbb{E}_{s \sim \rho(\cdot), (a^0, a^1) \sim \mu(\cdot|s)} \left[ x x^\top \right]$. We assume that $\lambda_{\min}(\Sigma) \geq \kappa$ for some $\kappa > 0$.

**Corollary 1.** *Under the same assumption in Theorem 1 and Assumption 2, with probability at least $1 - \alpha$, we have*

$$\left\| \hat{\theta}_{\text{RR}} - \theta^* \right\|_2 \leq O\left( \frac{1}{\gamma \kappa} \frac{e^{\varepsilon/m} + 1}{e^{\varepsilon/m} - 1} \sqrt{\frac{1 + \log(1/\alpha)}{nm}} \right)$$

*Remark* 1. The estimation error in Corollary 1 is influenced by the convergence parameter $\kappa$, which implicitly depends on the dimensionality $d$ (Wang et al., 2020). Since $\|x\| \leq L$, it follows that $\kappa \geq O\left( \frac{L^2}{d} \right)$ by Assumption 1. Thus, to implement user label level DP, the estimation error of estimator $\hat{\theta}_{\text{RR}}$ is $O\left( \frac{1}{e^{\varepsilon/m} - 1} \sqrt{\frac{d^2}{mn}} \right)$. When $m = 1$, i.e., when in the item-level case, our bound matches the nearly-optimal rate shown in Chowdhury et al. (2024b). However, when $m$ is large, $\frac{\varepsilon}{m}$ becomes a vanishingly small quantity. Consequently, the estimation error comes to $O\left( \frac{1}{\varepsilon} \sqrt{\frac{d^2 m}{n}} \right)$, which can be a large magnitude, rendering the estimator inefficient.

### 4.2 Lower Bound of User-level DP-RLHF

On the other hand, we introduce the lower bound of estimation error for user-level DP-RLHF, which characterizes the worst-case performance of any user-level DP algorithms. A detailed discussion is deferred to the Appendix B.5.

**Theorem 2** (Informal Statement). *For any $(\epsilon, \delta)$-user-level DP algorithm with output $\theta_{priv}$, there exists an instance of the BTL model with the underlying parameter $\theta^*$ such that*

$$\mathbb{E} \|\theta_{priv} - \theta^*\|_2 \geq \Omega \left( \frac{d}{\sqrt{nm}} + \frac{\sqrt{d}}{\sqrt{mn\epsilon}} \right). \tag{3}$$

## 5 Main Method

In the previous section, we have shown that although the RR-based mechanism for DP-RLHF in the user-level setting is simple, there are several issues. First, when the contribution

---

**Algorithm 1** AUP-RLHF

---

1: **Input:** Dataset $D = (Z_1, \ldots, Z_n) \in (\mathbb{Z}^m)^n$, privacy parameters $(\varepsilon, \delta)$, initial point $\theta_0$
2: Based on users, partition $D$ into $k$ disjoint datasets $\{D_i\}_{i \in [k]}$, where $D_i$ is of size $n_i :=$ $n/2^{k+1-i}$
3: **for** $i = 1, \ldots, k$ **do**
4:   Run Algorithm 2 with $(\theta_{i-1}, D_i, \tilde{n}_i, T_i, \eta_i, \varepsilon, \delta, \tau_i)$ and get its output $\theta_i$.
5: **end for**
6: **Output:** $\hat{\theta} = \theta_k$

---

**Algorithm 2** AdapUserPriv-SGD

---

1: **Input:** Initial model weights $\theta_0$, training set $\mathcal{D}$ of $n$ users and $m$ records, batch size $\tilde{n}$, learning rate $\eta$, iterations $T$, privacy parameters $(\varepsilon, \delta)$, concentration threshold $\tau$
2: $\varsigma \leftarrow \text{PrivacyAccount}(\frac{\varepsilon}{2}, \frac{\delta}{2}, n, \tilde{n}, T)$                     ▷ Compute noise multiplier
3: **for** $t = 1, \ldots, T$ **do**
4:   Randomly draw $U_t$, a batch of $\tilde{n}$ users                     ▷ Subsampling
5:   **for** $i \in U_t$ **do**
6:     $g_{t,i} \leftarrow \frac{1}{m} \sum_{j \in [m]} \nabla \ell(\theta_t, z_j)$                 ▷ Compute user-averaged gradient
7:   **end for**
8:   $s_t^c \leftarrow \frac{1}{\tilde{n}} \sum_{i,i' \in U_t} \mathbb{1}(\|g_{t,i} - g_{t,i'}\| \leq \tau)$                 ▷ Compute concentration scores
9:   **if** $\text{AboveThreshold}(s_t^c, \varepsilon/2, 4\tilde{n}/5) = \top$ in Alg. 3 **then**▷ Run AboveThreshold with $s_t^c$
10:     Set $B_t \leftarrow \varnothing$
11:     Set $f_{t,i} \leftarrow \sum_{i'} \mathbb{1}(\|g_{t,i'} - g_{t,i}\| \leq 2\tau)$
12:     Add $i$th user to $B_t$ with probability $p_{t,i}$, where                 ▷ Remove the outliers

$$p_{t,i} = \begin{cases} 0 & \text{if } f_{t,i} < \tilde{n}/2 \\ 1 & \text{if } f_{t,i} \geq 2\tilde{n}/3 \\ \frac{f_{t,i} - \tilde{n}/2}{\tilde{n}/6} & \text{otherwise} \end{cases}$$

13:     Aggregate and add noise for the remaining users:
14:     Let $\hat{g}_t = \frac{1}{|B_t|} \sum_{i \in B_t} g_{t,i}$ if $B_t$ is not empty, and 0 otherwise

15:     $\tilde{g}_t \leftarrow \hat{g}_t + \nu_t$, where $\nu_t \sim \mathcal{N}(0, \frac{8\tau^2 \log(e^\varepsilon T/\delta)\varsigma^2}{\tilde{n}^2})I_d$
16:     $\theta_{t+1} \leftarrow \theta_t - \eta \tilde{g}_t$                     ▷ Update model
17:   **else**
18:     Halt.                     ▷ Halt the algorithm if it does not pass.
19:   **end if**
20: **end for**
21: **return** $\hat{\theta} = \frac{1}{T} \sum_{t \in [T]} \theta_t$

---

of each user $m$ is large, we can see the privacy-utility trade-off in Corollary 1 is bad. This is because, intuitively, larger $m$ indicates we have more data, so the estimation error should be lower. However, a larger $m$ will introduce a larger error for the RR mechanism, which contradicts our expectations. Second, we can see there is a large gap between the lower bound and upper bound. Thus, a natural question is whether we can further fill in the gap. Finally, the private model $\hat{\theta}_{\text{RR}}$ needs to be the exact minimizer of the loss in (2), which is impossible to get in practice due to the non-convexity of the loss. Thus, a practical and efficient DP algorithm is needed.

In this section, we propose our AUP-RLHF method, which is based on DP-SGD (Abadi et al., 2016; Wang et al., 2017; Wang & Xu, 2019; Wang et al., 2019). Instead of flipping labels, here we consider the original loss function in the BTL model with linear reward:

$$\ell(\theta; z) := [\mathbb{1}(y_z = 1) \log p_{z,1} + \mathbb{1}(y_z = 0) \log p_{z,0}],$$

where $p_{z,1} = \sigma(x^\top \theta), p_{z,0} = (1 - \sigma(x^\top \theta))$.

In AUP-RLHF (Algorithm 1), we first partition the data into several disjoint sets, and for each set, we will use a DP-SGD-based update, namely AdapUserPriv-SGD (Algorithm 2). In detail, during each iteration of parameter updates, the framework operates in three stages. First, a subset of users is selected via user-wise sampling, and the average gradient of the selected subset is computed as a query (Step 4-8). Second, the query is passed through a user-level private mean estimation oracle, which outputs a gradient satisfying user-level DP (Step 9-15). Third, the gradient obtained from the previous step is used to perform gradient descent for parameter updates.

The main difference between AUP-RLHF and DP-SGD is the private mean estimation oracle for aggregated gradients. If we directly extend DP-SGD to the user-level (Appendix Algorithm 4) (Chua et al., 2024) , for each user's data, we may aggregate their gradient and perform a clipping with some threshold $C$ (Step 9). In this case, the sensitivity of the average of clipped gradients is $O(\frac{1}{\tilde{n}})$, with $\tilde{n}$ as the number of subsampled users. However, such a method does not leverage information on each user's data, leading to unsatisfactory performance. In this work, we leverage the user-level private mean estimation oracle proposed by Liu & Asi (2024b), which is an adaptive sampling procedure to remove outliers to let the aggregated user-level gradients add less noise. To achieve adaptive mean estimation for concentrated samples, we first introduce the concept of concentration.

**Definition 3.** A set of random samples $\{X_i\}_{i \in [n]}$ is $(\tau, \gamma)$-concentrated if there exists a point $x \in \mathbb{R}^d$ such that with probability at least $1 - \gamma$,

$$\max_{i \in [n]} \|X_i - x\| \le \tau.$$

After adaptive sampling (Step 8-14), we can obtain a $\tau$-concentrated subset of users, with user-averaged gradients $\{g_{t,i}\}_{i \in U_t}$ as the random samples. Due to the concentration property, the sensitivity of the aggregated gradients will be bounded $O(\tau)$. Thus, there is no need to do a clipping and it allows the added noise to scale proportionally to $\tau$ (which is $\tilde{O}(\frac{1}{\sqrt{m}})$) rather than the constant clipping radius $C$, resulting in significantly smaller noise and achieving improved utility.

Specifically, to get such a concentrated subset, the algorithm first employs an *AboveThreshold* mechanism (Algorithm 3) to compute a concentration score, which determines whether the average gradient of users is predominantly concentrated (Step 9). This ensures that the input dataset is approximately $\tau$-concentrated. Subsequently, outlier detection is performed by assigning each sample a score that quantifies its likelihood of being an outlier, i.e., not concentrated gradient. Each sample is then retained with a probability proportional to its score, effectively removing low-score outliers and yielding a high-quality, concentrated subset of data (Step 10-12).

**PrivacyAccount.** Given the total number of users, batch size of users, number of iterations, and privacy budget, this subroutine can determine the noise $\varsigma$ we need to add in Step 16 of Algorithm 2 to ensure the algorithm satisfies $(\varepsilon, \delta)$ DP. We adopt the Poisson subsampling, a common practice in the privacy accounting of DP-SGD, in Step 4. Thus, theoretically, we can show that $\varsigma = \tilde{O}(\frac{\varepsilon n}{\sqrt{T}\tilde{n}})$ can make the algorithm achieve $(\varepsilon, \delta)$-DP (3). However, in practice, we always use various open source libraries to give more accurate calculations on such a noise, such as DP Accounting Library[1].

**Theorem 3.** *For any $0 < \varepsilon, \delta < 1$, if $\varsigma = \tilde{O}(\frac{n}{\tilde{n}} \frac{\varepsilon}{\sqrt{T \ln(1/\delta)}})$, Algorithm 1 is $(\varepsilon, \delta)$ user-level (label) DP. Here, the Big-$\tilde{O}$ notation omits other logarithmic terms.*

In the following, we will show that it is possible to get an improved upper bound of estimation error with some specific hyper-parameters.

---

[1] https://github.com/google/differential-privacy/tree/main/python/dp_accounting

**Theorem 4.** *Under Assumption 1 and 2 and the condition in Theorem 3, for any $0 \leq \varepsilon \leq 10$ and $0 \leq \delta \leq 1$, in Algorithm 1 set $k = \lceil \log\log mn \rceil$, $\theta_0 = 0$, $\tilde{n}_i = n_i$, $\eta_i = \frac{B}{4L} \cdot$ $\min\left\{ \frac{\sqrt{m}n_i\varepsilon}{T_i\sqrt{d\log^2(mn_id/\delta)}}, \frac{1}{T_i^{3/4}}, \frac{\sqrt{n_im}}{T_i} \right\}$, $\tau_i = \frac{4L\log(n_idme^\varepsilon T_i/\delta)}{\sqrt{m}}$, and $T_i = O\left(m^2n_i^2 + mn_i\sqrt{d}\right)$. The output $\hat{\theta}$ satisfies the following if $n_i > \frac{\log(mdn_i)\log(mdn_i/\delta)}{\varepsilon}$,*

$$\mathbb{E}\|\hat{\theta} - \theta^*\|_2 \leq \widetilde{O}(\frac{L}{\kappa\gamma}\left[\frac{1}{\sqrt{mn}} + \frac{\sqrt{d}}{\sqrt{mn}\varepsilon}\right]).$$

*The Big-$\widetilde{O}$ notation omits other logarithmic terms.*

*Remark 2.* We discuss the case under the same condition as in Remark 1 when $\kappa = O\left(\frac{L^2}{d}\right)$. As we mentioned, when $m$ is a large quantity, the estimation error of the RR estimator is $O(\frac{1}{\varepsilon}\sqrt{\frac{d^2m}{n}})$, while the upper bound of AUP-RLHF is $\widetilde{O}\left(\frac{d\sqrt{d}}{\sqrt{mn}\varepsilon}\right)$. We can see it improves a factor of $O(\frac{m\sqrt{n}}{\sqrt{d}})$. The factor of $O(\frac{1}{\sqrt{d}})$ is because our SGD algorithm adds noise to each dimension of the parameter $\theta$. When $m\sqrt{n}$ exceeds $\sqrt{d}$, our algorithm exhibits better utility than the RR algorithm, which is also commonly observed in practical scenarios.

In the previous section, we considered the case where the label is binary. In fact, our method can be directly extended to the general *K*-wise case. See Appendix B.4 for details.

# 6 Experiments

## 6.1 Experimental Setup

**Tasks and Datasets.** We evaluate our AUP-RLHF method on two different datasets with three different tasks. The IMDb dataset (Maas et al., 2011) is a widely used movie review dataset containing both positive and negative reviews. We perform a controlled sentiment generation task on this dataset. The TL;DR dataset (Völske et al., 2017) is a summarization dataset containing posts from Reddit along with their corresponding titles, on which we conduct a summarization task.

**Base models.** For each dataset and task, we use the pre-trained Gemma-2-2B (Team et al., 2024) and LLaMA-2-7B (Touvron et al., 2023) as our base models.

**Baselines.** We compare our algorithm with three baselines, which all are user-level DP-based RLHF algorithms: Random Response (Section 4), User-wise DP-SGD (Chua et al., 2024), and Group Privacy (Chua et al., 2024). Additionally, we compare our results with those obtained from RLHF under non-private settings.

**Evaluation.** In the controlled sentiment generation task, we use the reward score as the evaluation metric. Additional details are provided in Appendix A.1. For the summarization task, we assess our AUP-RLHF algorithm by comparing its win rate against each baseline strategy. Following previous work in RLHF (Rafailov et al., 2024), we adopt GPT-4 as a proxy for human evaluation.

**Settings.** We set each user's contributions at $m = 10$, with $n$ set to 2500 for the sentiment generation task, and $m = 50, n = 1800$ for the summarization task. We set privacy budgets as $\varepsilon = 3$ or $\varepsilon = 8$, with $\delta$ fixed at $10^{-5}$. Due to the slower training speed of the SGD-based optimizer compared to Adam, we set the number of sub-datasets $k = 1$ and the number of training epochs for the reward model to 5 instead of 1. Other hyperparameter details are provided in Appendix A Table 3 and Table 4. The experiments are carried out on machines with one Nvidia A100 GPU card, 14-core Intel Xeon Gold 6348 CPUs, and 100GB of RAM.

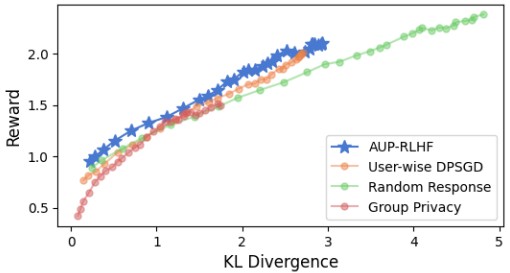

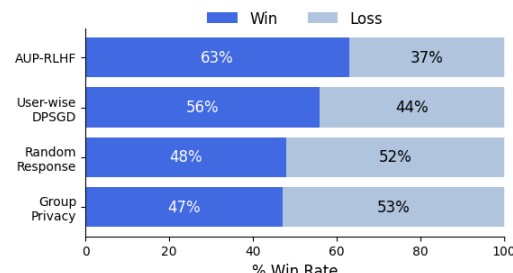

Figure 1: IMDb Sentiment Generation (Llama-2-7b, $\epsilon = 8$).

Figure 2: Win Rate Against the SFT Model for TL;DR Summarization (Gemma-2-2b, $\epsilon = 8$).

## 6.2 Experimental Results

**IMDb Sentiment.** For the sentiment generation task, we make the following observations based on the experimental results depicted in Figure 1, 3, and 4 : (1) We observe that, under the same KL divergence, our AUP-RLHF outperforms other baselines in terms of the reward metric, demonstrating the superiority of our model. (2) Consistent with the theoretical analysis of differential privacy, relaxing the privacy budget leads to an improvement in the reward score metric. Specifically, we find that, for the same KL divergence, the reward values of AUP-RLHF and all baselines are higher when $\epsilon = 8$ compared to when $\epsilon = 3$. (3) By comparing the experimental results of Gemma-2-2B and Llama-2-7B, we observe that at $\epsilon = 3$, the larger model yields improved performance, which is consistent with most DP fine-tuning papers (Yu et al., 2021; Wu et al., 2023a). However, at $\epsilon = 8$, the performance improvement in the larger model is less significant, possibly due to the lack of a sufficiently exhaustive hyperparameter search to find the optimal model performance.

**TL;DR Summarization.** Similar to the findings in the sentiment generation task, in Figure 2 and 5 we observe that AUP-RLHF outperforms other baselines in terms of the win rate metric, across both model sizes and varying privacy budgets. Furthermore, when the privacy budget is set to $\epsilon = 3$, the performance of all algorithms is suboptimal, with winrates below 50%. However, when the privacy budget is relaxed to $\epsilon = 8$, the win rates of all algorithms improve. Meanwhile, our AUP-RLHF achieves a significant increase in winrate, rising from 39% to 63% on the Gemma-2-2B model.

| $m$ | 5 | 10 | 20 | 50 |
|---|---|---|---|---|
| $\epsilon = 1$ | 0.366 | 0.586 | 1.051 | 2.429 |
| $\epsilon = 3$ | 0.139 | 0.175 | 0.246 | 0.461 |
| $\epsilon = 8$ | 0.086 | 0.102 | 0.128 | 0.189 |

Table 1: Effective noise of AUP-RLHF on TL;DR dataset.

| $m$ | 5 | 10 | 20 | 50 |
|---|---|---|---|---|
| $\epsilon = 1$ | 1.101 | 1.284 | 1.631 | 2.402 |
| $\epsilon = 3$ | 0.726 | 0.809 | 0.928 | 1.182 |
| $\epsilon = 8$ | 0.531 | 0.578 | 0.639 | 0.750 |

Table 2: Effective noise of User-wise DPSGD on TL;DR dataset.

**Analyzing AUP-RLHF.** As observed in Table 1, as the number of user records $m$ increases, the noise added to AUP-RLHF increases accordingly. This is because, based on our theory, the variance of the noise is $\tilde{O}(\frac{1}{m\tilde{n}^2})$. Thus, when the total size $mn$ and iteration number $T$ are fixed, $m\tilde{n} = \frac{mn}{T}$ is fixed, so increasing $m$ reduces $\tilde{n}$, making the noise $\tilde{O}(\frac{1}{m\tilde{n}^2})$ larger.

Additionally, we find that, under the same $\epsilon$ and $m$, the noise in AUP-RLHF is smaller than that in User-wise DP-SGD. This is due to the fact that our algorithm, based on the user-concentrated nature, requires the noise scale to be proportional to the concentration parameter $\tau$ rather than the clipping radius $C$. This reduction in noise allows our algorithm to converge more quickly.

# 7 Conclusion

In this work, we study RLHF with user-level privacy. We first demonstrate the sub-optimality of the RR method, followed by presenting the lower bound for user-level DP-RLHF. Then, we propose the AUP-RLHF algorithm, which ensures that the output satisfies user-label DP while achieving better utility. In our experiments, we validate the superiority of our algorithm across two commonly used datasets.

Although we present an innovative approach, there are some limitations to our method that motivate future research directions. First, our upper bound (Theorem 4) relies on a strong assumption that the loss function is both strongly convex and Lipschitz continuous. Second, this upper bound we prove is suboptimal, which has an additional factor of $d$ compared to lower bound. This originates from our coverage assumption. Furthermore, when extending our analytical framework to DPO, a non-linear assumption is required (Chowdhury et al., 2024a; Zhu et al., 2023), making it difficult to unify the semi-norm $\|\cdot\|_{\Sigma_D}$ with the $l_2$-norm.

## Acknowledgements

Di Wang and Zihang Xiang are supported in part by the funding BAS/1/1689-01-01, URF/1/5508-01-01 from KAUST, and funding from KAUST - Center of Excellence for Generative AI, under award number 5940.

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

# A Additional Experiment Details

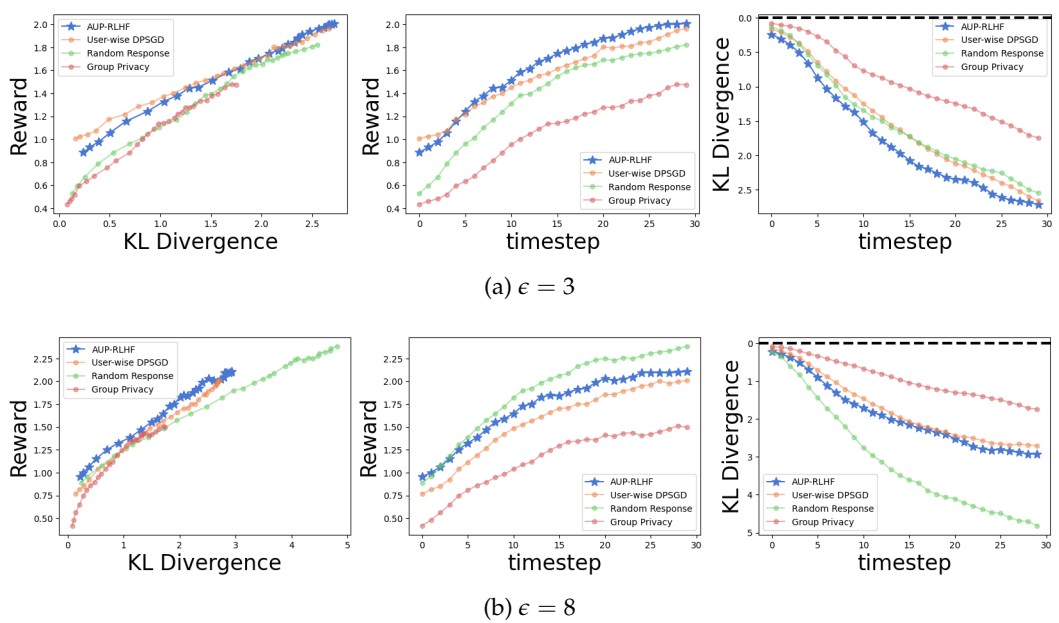

Figure 3: IMDb Sentiment Generation (Llama-2-7b).

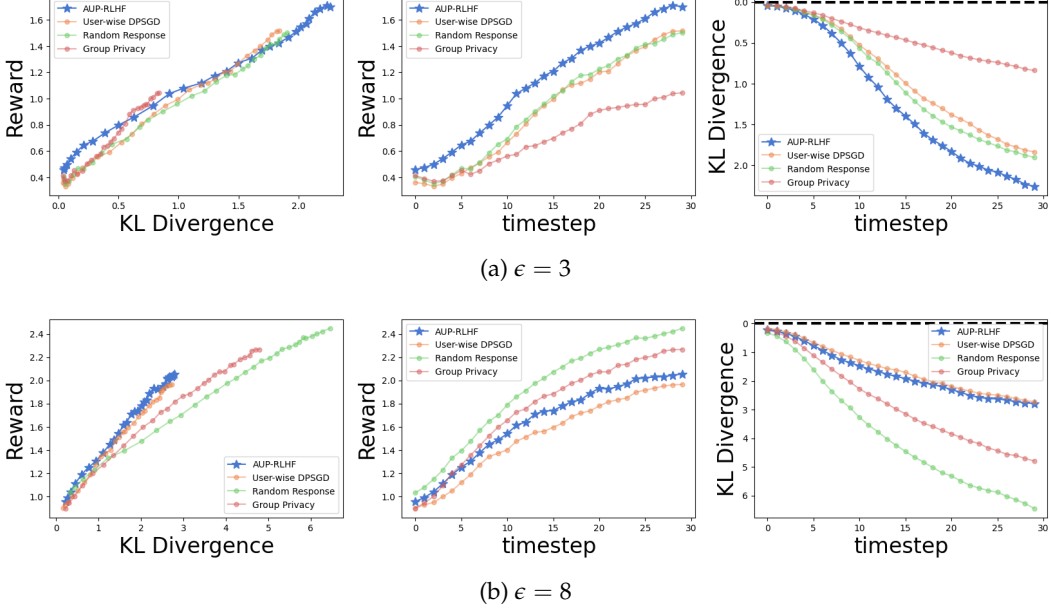

Figure 4: IMDb Sentiment Generation (Gemma-2-2b).

## A.1 IMDb Sentiment Experiment Details

We follow the setup in Rafailov et al. (2024), where prefix prompts are sampled between 2 to 8 tokens. For reward model training, we use `cardiffnlp/twitter-roberta-base-sentiment`, a backbone model trained on Twitter data, which is further fine-tuned on IMDb sentiment analysis by updating the last layer only. Llama-2-7b and Gemma-2-2b serve as the base models. While each method fine-tunes the base models using its respective reward model, test performance is evaluated using a ground-truth reward model, `siebert/sentiment-roberta-large-english`, ensuring a fair comparison. On every 10 steps until convergence, we report the rewards and KL-divergence over a set of test prompts, shown in Fig 3 and Fig 4.

| Parameter | Gemma-2-2B | Llama-2-7B |
|---|---|---|
| Batch Size | 4 | 4 |
| Learning Rate | 1e-4 | 1e-4 |
| Warmup Ratios | 0.1 | 0.1 |
| Learning Rate Scheduler | Cosine | Cosine |
| Optimizer | AdamW | AdamW |
| Training Epochs | 1 | 1 |
| LoRA Rank | 16 | 16 |
| LoRA Alpha | 32 | 32 |

Table 3: Parameter settings for Supervised Fine-Tuning (SFT) on Gemma-2-2B and Llama-2-7B.

| Parameter | Gemma-2B | Llama-2-7B |
|---|---|---|
| Batch Size | 64 | 64 |
| Mini-batch Size | 64 | 64 |
| Learning Rate | 1e-5 | 1e-5 |
| Optimizer | RMSProp | RMSProp |
| KL Penalty Coefficient | 0.15 | 0.15 |
| LoRA Rank | 16 | 16 |
| LoRA Alpha | 32 | 32 |

Table 4: PPO parameter settings for Gemma-2-2B and Llama-2-7B.

| Parameter | AUP-RLHF | User-wise DPSGD | Group Privacy | Random Response |
|---|---|---|---|---|
| User Batch Size | 50 | 50 | N/A | N/A |
| Per-user Sample Size | 10 | 10 | N/A | N/A |
| Sample Batch Size | N/A | N/A | 500 | 500 |
| Learning Rate | 1e-3 | 1e-3 | 1e-3 | 1e-3 |
| Warmup Ratios | 0.2 | 0.2 | 0.2 | 0.2 |
| Learning Rate Scheduler | Cosine | Cosine | Cosine | Cosine |
| Epochs | 5 | 5 | 5 | 5 |

Table 5: Reward model parameter settings for Gemma-2-2B and Llama-2-7B.

## A.2 TL;DR Summarization

For reward model training, we use `tasksource/ModernBERT-base-nli`, a backbone model trained on Natural Language Inference, which is further fine-tuned on summarization preference by updating the last layer only.

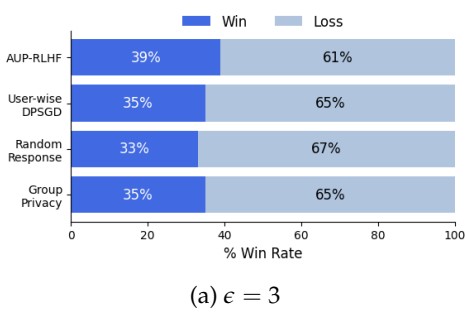 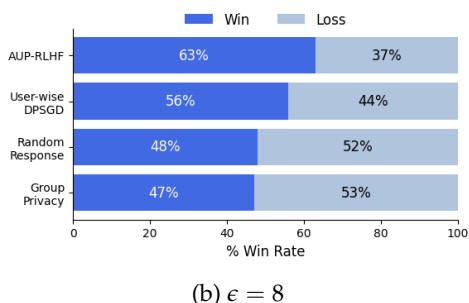

(a) $\epsilon = 3$        (b) $\epsilon = 8$

Figure 5: Win Rate Against the SFT Model for TL;DR Summarization (Gemma-2-2b).

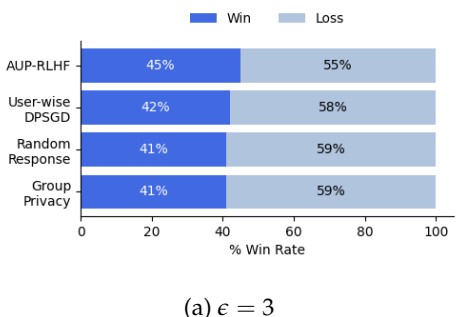 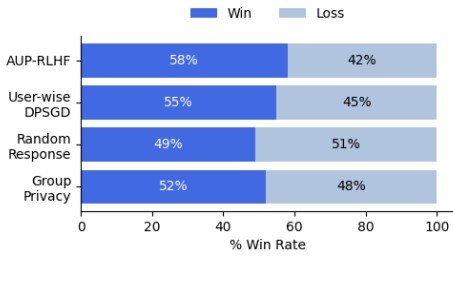

(a) $\epsilon = 3$        (b) $\epsilon = 8$

Figure 6: Win Rate Against the SFT Model for TL;DR Summarization (Llama-2-7b).

### A.3 Computational overhead

The main overhead of group privacy training methods is introduced by the per-example gradient clipping and noise addition. For the user-wise DP-SGD and AUP-RLHF, the main overhead is from the user-wise data sampling(line 4 in Algorithm 2), per-user gradient clipping(line 9 in Algorithm 4) or concentration scores computing(line 8 in Algorithm 2) and noise addition(line 15 in Algorithm 2)

We report the average per-step training time (in seconds) for training the reward model using different methods, averaged over 100 steps 100 steps with a batch size of 500 on the IMDb Sentiment Generation task.

|  | Random Response | Group Privacy | User-wise DPSGD | AUP |
|---|---|---|---|---|
| Training Time | 2.78 | 4.29 | 4.32 | 4.54 |

Table 6: Training time per step (seconds)

From the Table 6, we observe that AUP-RLHF incurs only a little computational overhead compared to Group Privacy and User-wise DP-SGD. The relatively short training time of Random Response is due to the absence of gradient clipping and noise addition.

### A.4 Number of contributions per user

According to Theorem 5, when the number of users $n$ is fixed, a larger $m$ results in a smaller upper bound on the estimation error, leading to better utility.

To provide a more quantitative analysis, we present the following table showcasing the effect of varying . We conduct experiments on IMDb Sentiment Generation using Gemma-2 2B, following the same settings as in Tables 3, 4, and 5.

| m | 10 | 100 | 250 | 500 |
|---|---|---|---|---|
| Reward | 0.6402 | 0.8582 | 0.8539 | 1.2671 |
| KL divergence | 0.6960 | 0.9071 | 0.7434 | 0.8800 |
| Reward / KL divergence | 0.9198 | 0.9461 | 1.1486 | 1.4399 |

Table 7: Effect of contribution number $m$

As shown in the Table 7, We observe that as the user contribution $m$ increases, the reward-to-KL divergence ratio improves at convergence, indicating that the model achieves better utility.

## B    Other Details

### B.1    Notions

| Parameter | Meaning | Parameter | Meaning |
|---|---|---|---|
| $n$ | Number of users | $m$ | Contributions per user |
| $\theta$ | Reward model parameter | $\theta^*$ | Ground-truth reward model parameter |
| $d$ | Dimension of $\theta$ | $\varepsilon$ | Privacy budget |
| $\delta$ | Failure probability | $K$ | Number of candidate responses |
| $\sigma$ | Sigmoid function | $r_\theta$ | Reward function parameterized by $\theta$ |
| $\phi$ | Feature map | $x$ | Differential feature vector |
| $y$ | Human preference label | $\tilde{y}$ | Randomized label |
| $\Sigma$ | Covariance matrix | $\lambda_{\min}$ | Minimum eigenvalue |
| $\gamma$ | Strong convexity parameter | $L$ | Lipschitz constant |
| $B$ | Bound on $|\theta|$ | $\kappa$ | Coverage parameter |
| $\tau$ | Concentration threshold | $C$ | Gradient clipping bound |
| $\alpha$ | Probability bound | $\eta$ | Learning rate |
| $T$ | Number of iterations | $\bar{n}$ | Batch size of users |
| $\mathcal{D}$ | Dataset | $\mathcal{P}$ | Distribution family |
| $\varsigma$ | Noise multiplier | $\Delta$ | Parameter difference |

Table 8: Key notions in this paper

### B.2    Algorithm

---

**Algorithm 3** AboveThreshold

---

1: **Input:** Dataset $\mathcal{D} = (Z_1, \ldots, Z_n)$, threshold $\Delta \in \mathbb{R}$, privacy parameter $\varepsilon$
2: Let $\hat{\Delta} := \Delta - \text{Lap}\left(\frac{2}{\varepsilon}\right)$
3: **for** $t = 1$ **to** $T$ **do**
4:     Receive a new query $q_t : \mathbb{Z}^n \to \mathbb{R}$
5:     Sample $\nu_t \sim \text{Lap}\left(\frac{4}{\varepsilon}\right)$
6:     **if** $q_t(\mathcal{D}) + \nu_t < \hat{\Delta}$ **then**
7:         Output: $a_t = \bot$
8:         Halt
9:     **else**
10:         Output: $a_t = \top$
11:     **end if**
12: **end for**

---

---

**Algorithm 4** User-wise DP-SGD

---

1: **Input:** Initial model weights $\theta_0$, training set $D$ of $n$ users and $m$ records, learning rate $\eta$, iterations $T$, user batch size $\tilde{n}$, privacy budget $\varepsilon$, number of records per user $\{k_i\}_{i=1}^{\tilde{n}}$ gradient norm bound $C$

2: $\varsigma \leftarrow \text{PRIVACYCOUNTING}(\varepsilon, \delta, n, \tilde{n}, T)$          ▷ Compute noise multiplier

3: (User-wise DP-SGD directly uses $D$)          ▷ Prepare the dataset

4: **for** $t = 1, \ldots, T$ **do**

5:     Randomly draw $U_t$, a batch of $\tilde{n}$ users

6:     **for** $i = 1, \ldots, \tilde{n}$ **do**

7:         Sample $z_j = \{x_j, y_j\}_{j \in [k_i]}$ $k_i$ records for the $i$th user in $U_t$

8:         $g_{t,i} \leftarrow \frac{1}{k_i} \sum_{j \in [k_i]} \nabla_{\theta_t} \ell(\theta, z_j)$         ▷ Compute user-averaged gradient

9:         $\hat{g}_{t,i} \leftarrow g_{t,i} / \max(1, \frac{\|g_{t,i}\|_2}{C})$         ▷ Clip gradient

10:     **end for**

11:     $\tilde{g}_t \leftarrow \frac{1}{\tilde{n}} \sum_{j \in [\tilde{n}]} \hat{g}_{t,i} + \mathcal{N}(0, \varsigma^2 C^2 I_d)$         ▷ Aggregate and add noise

12:     $\theta_{t+1} \leftarrow \theta_t - \eta \tilde{g}_t$         ▷ Model update

13: **end for**

---

### B.3 De-biased loss function

We start with the BCE loss under randomized labels. Next, we discuss their drawback, which help us get intuition for a de-biased loss. For any $\theta \in \mathbb{R}^d$, the predicted probabilities of a randomized label $\tilde{y}_{i,j}$ given $x_{i,j}$ as

$$\tilde{p}_{i,j}^1 = \sigma(\varepsilon)\sigma\left(x_{i,j}^\top \theta\right) + (1 - \sigma(\varepsilon))\left(1 - \sigma\left(x_{i,j}^\top \theta\right)\right),$$
$$\tilde{p}_{i,j}^0 = \sigma(\varepsilon)\left(1 - \sigma\left(x_{i,j}^\top \theta\right)\right) + (1 - \sigma(\varepsilon))\sigma\left(x_{i,j}^\top \theta\right).$$

After applying random response to the dataset, it becomes $\tilde{\mathcal{D}} = \left((x_{i,j}, \tilde{y}_{i,j})\right)_{j=1}^m)_{i=1}^n$. The MLE $\tilde{\theta}_{\text{MLE}}$ computed on this dataset satisfies user-level privacy and can be obtained by minimizing the BCE loss (the negative log-likelihood).

$$l_{\mathcal{D},\varepsilon}(\theta) = -\sum_{i=1}^n \sum_{j=1}^m \left[\mathbb{1}\left(\tilde{y}_{i,j} = 1\right) \log \tilde{p}_{i,j}^1 + \mathbb{1}\left(\tilde{y}_{i,j} = 0\right) \log \tilde{p}_{i,j}^0\right].$$

The drawback of BCE loss with randomized labels is that it introduces bias in loss compared with clear-text loss $\mathbb{E}[l_{\mathcal{D},\varepsilon}(\theta)] \neq \mathbb{E}[l_{\mathcal{D}}(\theta)]$. At the same time, this leads to a discrepancy in the log-odds between the randomized dataset and the clear dataset, introducing bias in the preferences for $a^1$ and $a^0$ (Chowdhury et al., 2024b). This motivates us to design a de-bias loss function that ensures the log-odds are identical between the randomized and clear datasets. The following loss achieves this:

$$\hat{l}_{\mathcal{D},\varepsilon}(\theta) = -\sum_{i=1}^n \sum_{j=1}^m \left[\mathbb{1}\left(\tilde{y}_{i,j} = 1\right) \log \hat{p}_{i,j}^1 + \mathbb{1}\left(\tilde{y}_{i,j} = 0\right) \log \hat{p}_{i,j}^0\right]$$

where we define, for any $\theta \in \mathbb{R}^d$, the predicted scores of each randomized label $\tilde{y}_{i,j}$ given $x_{i,j}$ as

$$\hat{p}_{i,j}^1 = \frac{\sigma\left(x_{i,j}^\top \theta\right)^{\sigma(\varepsilon)}}{\left(1 - \sigma\left(x_{i,j}^\top \theta\right)\right)^{(1-\sigma(\varepsilon))}}, \quad \hat{p}_{i,j}^0 = \frac{\left(1 - \sigma\left(x_{i,j}^\top \theta\right)\right)^{\sigma(\varepsilon)}}{\sigma\left(x_{i,j}^\top \theta\right)^{(1-\sigma(\varepsilon))}}.$$

Although $\hat{p}_{i,j}^1$ and $\hat{p}_{i,j}^0$ are not probabilities, they satisfy our desired property:

$$\log \frac{\widehat{p}_{i,j}^1}{\widehat{p}_{i,j}^0} = \log \frac{\sigma\left(x_{i,j}^\top \theta\right)}{1 - \sigma\left(x_{i,j}^\top \theta\right)} = \text{logit}\left(p_{i,j}^1\right).$$

Hence, the loss function $\widehat{l}_{\mathcal{D},\varepsilon}(\theta)$ essentially de-biases the effect of randomization.

### B.4 Extension to K-wise

Let $(s, a^1, ..., a^K, y)$ be the data structure of K-wise RLHF, where each sample has a state $s \in \mathcal{S}$ (e.g., prompt given to a language model), K actions $a^1, ..., a^K \in \mathcal{A}$ (e.g., K responses from the language model), and label $y \in \{1, 2, ..., K\}$ indicating which action is most preferred by human experts. We assume that the state $s$ is first sampled from some fixed distribution $\rho$. The K actions $(a^1, ..., a^K)$ are then sampled from some joint distribution (i.e., a behavior policy) $\mu$ conditioned on $s$. Under the Plackett-Luce model (Plackett, 1975; Luce, 1959), the label $y$ is sampled according to the probability distribution:

$$\mathbb{P}_{\theta^*}[y = k \mid s, a_1, \ldots, a_K] = \frac{\exp(r_{\theta^*}(s, a_k))}{\sum_{j=1}^K \exp\left(r_{\theta^*}\left(s, a_j\right)\right)}$$

For the Plackett-Luce (PL) model Plackett (1975); Luce (1959) for $K$-wise comparisons between actions, let $\Pi$ be the set of all permutations $\pi : [K] \to [K]$, which denotes the ranking of $K$ actions, where $a_{\pi(j)}$ denotes the $j$-th ranked action. Under the PL model, we define the loss of a permutation $\pi \in \Pi$ for a state $s$ as

$$\ell(\theta; s, \pi) = -\log\left(\prod_{j=1}^K \frac{\exp\left(r_\theta\left(s, a_{\pi(j)}\right)\right)}{\sum_{k'=j}^K \exp\left(r_\theta\left(s, a_{\pi(k')}\right)\right)}\right).$$

**Theorem 5.** *Under the same setting of Theorem 4, the output $\hat{\theta}$ of Algorithm 1 with loss function $\ell(\theta; s, \pi)$ satisfies*

$$\mathbb{E}\|\hat{\theta} - \theta^*\| \leq \widetilde{O}\left(\frac{K^2 L}{\kappa \gamma}\left[\frac{1}{\sqrt{mn}} + \frac{\sqrt{d}}{\sqrt{mn\varepsilon}}\right]\right)$$

*The Big-$\widetilde{O}$ notation omits other logarithmic terms.*

*Remark 3.* When $K = 2$ or $K = O(1)$, the PL model reduces to the pairwise comparison considered in the BTL model, yielding the same utility as in Theorem 4.

### B.5 Lower Bound of User-level DP-RLHF

On the other side, we will study the lower bound of the estimation error for user-level DP-RLHF. Consider a family of distributions $\mathcal{P}$ over $(\mathcal{X}^m)^n$, where $\mathcal{X}$ represents the data universe, $m$ denotes the sample size, and $n$ refers to the user size. Our goal is to estimate a parameter $\theta$, which is a mapping $\theta : \mathcal{P} \to \theta$ that characterizes the underlying distribution. To quantify the estimation error, we define a pseudo-metric $\rho : \Theta \times \Theta \to \mathbb{R}_+$, which serves as the loss function for evaluating the accuracy of the estimate. The minimax risk under the loss function $\rho$ for the class $\mathcal{P}$ is given by:

$$R(\mathcal{P}, \rho) := \min_{\hat{\theta}} \max_{P \in \mathcal{P}} \mathbb{E}_{X \sim P}[\rho(\hat{\theta}(X), \theta(P))].$$

In the following, we focus on estimating an unknown parameter $\theta$, within the BTL model under a use-level privacy framework. Data is collected from $n$ users, where each user provides $m$ samples. We consider a fixed design setup, meaning that the feature vectors $x_{i,j} \in \mathbb{R}^d$ for samples $i \in [n]$ from user $j \in [m]$ are known. Our objective is to infer $\theta$ based on a sequence of private responses $\tilde{y}_{i,j}$.

**Theorem 6.** *For a large enough $n, m$, any private estimator $\widehat{\theta}$ based on samples form the BTL model that satisfies $(\varepsilon, \delta)$-user-level-label DP has the estimation error lower bounded as*

$$\mathbb{E}\left[\left\|\widehat{\theta} - \theta^*\right\|\right] \geq \Omega\left(\frac{d}{\sqrt{nm}} + \frac{\sqrt{d}}{\sqrt{mn}(\varepsilon + \delta)}\right).$$

*Remark* 4. The estimation bound in Theorem 6 consists of two components: a non-private term and a private term. Compared to the item-level setting in Chowdhury et al. (2024b), both the non-private and the private terms include an additional factor of $1/\sqrt{m}$. In particular, when $m = 1$, our results reduce to those in Chowdhury et al. (2024b), demonstrating consistency and generality with their findings.

## C   Omitted Proofs

### C.1   Proof of Theorem 1

*Proof.* We follow the proof method of Theorem 4.2 in Chowdhury et al. (2024b) to prove the sub-optimality of the random response algorithm at user-level setting.
First recall from our de-biased loss function 1

$$\hat{l}_{D,\varepsilon}(\theta) = -\frac{1}{nm}\sum_{i=1}^{n}\sum_{j=1}^{m}\left[1(\tilde{y}_{i,j} = 1)\left(\sigma(\varepsilon/m)\log\sigma(\theta^{\top}x_{i,j}) - (1 - \sigma(\varepsilon/m))\log(1 - \sigma(\theta^{\top}x_{i,j}))\right)\right.$$

$$\left. + 1(\tilde{y}_{i,j} = 0)\left(\sigma(\varepsilon/m)\log(1 - \sigma(\theta^{\top}x_{i,j})) - (1 - \sigma(\varepsilon/m))\log\sigma(\theta^{\top}x_{i,j})\right)\right].$$

The gradient of the loss function is given by

$$\nabla\hat{l}_{D,\varepsilon}(\theta) = -\frac{1}{nm}\sum_{i=1}^{n}\sum_{j=1}^{m}V_{\theta,i,j}\cdot x_{i,j} = -\frac{1}{nm}X^{\top}V_{\theta},$$

where

$$V_{\theta,i,j} = 1(\tilde{y}_{i,j} = 1)\left(\frac{\sigma'(\theta^{\top}x_{i,j})}{\sigma(\theta^{\top}x_{i,j})}\sigma\left(\frac{\varepsilon}{m}\right) + \frac{\sigma'(\theta^{\top}x_{i,j})}{1 - \sigma(\theta^{\top}x_{i,j})}\left(1 - \sigma\left(\frac{\varepsilon}{m}\right)\right)\right)$$

$$- 1(\tilde{y}_{i,j} = 0)\left(\frac{\sigma'(\theta^{\top}x_{i,j})}{1 - \sigma(\theta^{\top}x_{i,j})}\sigma\left(\frac{\varepsilon}{m}\right) + \frac{\sigma'(\theta^{\top}x_{i,j})}{\sigma(\theta^{\top}x_{i,j})}\left(1 - \sigma\left(\frac{\varepsilon}{m}\right)\right)\right).$$

It holds that

$$\mathbb{E}_{\theta}[V_{\theta,i,j}|x_{i,j}] = \left(\sigma(\theta^{\top}x_{i,j})\sigma(\varepsilon/m) + (1 - \sigma(\theta^{\top}x_{i,j}))(1 - \sigma(\varepsilon/m))\right)\left(\frac{\sigma'(\theta^{\top}x_{i,j})}{\sigma(\theta^{\top}x_{i,j})}\sigma(\varepsilon/m) + \frac{\sigma'(\theta^{\top}x_{i,j})}{1 - \sigma(\theta^{\top}x_{i,j})}(1 - \sigma(\varepsilon/m))\right)$$

$$- \left((1 - \sigma(\theta^{\top}x_{i,j}))\sigma(\varepsilon/m) + \sigma(\theta^{\top}x_{i,j})(1 - \sigma(\varepsilon/m))\right)\left(\frac{\sigma'(\theta^{\top}x_{i,j})}{1 - \sigma(\theta^{\top}x_{i,j})}\sigma(\varepsilon/m) + \frac{\sigma'(\theta^{\top}x_{i,j})}{\sigma(\theta^{\top}x_{i,j})}(1 - \sigma(\varepsilon/m))\right) = 0.$$

Furthermore, we have

$$|V_{\theta,i,j}|_{\tilde{y}_{i,j}=1} = \frac{\sigma'(\theta^{\top}x_{i,j})}{\sigma(\theta^{\top}x_{i,j})}\sigma(\varepsilon/m) + \frac{\sigma'(\theta^{\top}x_{i,j})}{1 - \sigma(\theta^{\top}x_{i,j})}(1 - \sigma(\varepsilon/m)),$$

$$|V_{\theta,i,j}|_{\tilde{y}_{i,j}=0} = \frac{\sigma'(\theta^{\top}x_{i,j})}{1 - \sigma(\theta^{\top}x_{i,j})}\sigma(\varepsilon/m) + \frac{\sigma'(\theta^{\top}x_{i,j})}{\sigma(\theta^{\top}x_{i,j})}(1 - \sigma(\varepsilon/m)).$$

The first derivative of the logistic function $\sigma(\cdot)$ is given by $\sigma'(z) = \sigma(z)(1 - \sigma(z))$, which gives us

$$|V_{\theta,i,j}|_{\tilde{y}_{i,j}=1} = (1 - \sigma(\theta^{\top}x_{i,j}))\sigma(\varepsilon/m) + \sigma(\theta^{\top}x_{i,j})(1 - \sigma(\varepsilon/m)) = \mathbb{P}_{\theta}[\tilde{y}_{i,j} = 0|x_{i,j}]$$

$$|V_{\theta,i,j}|_{\tilde{y}_{i,j}=0} = \sigma(\theta^\top x_{i,j})\sigma(\varepsilon/m) + (1 - \sigma(\theta^\top x_{i,j}))(1 - \sigma(\varepsilon/m)) = \mathbb{P}_\theta[\tilde{y}_{i,j} = 1|x_{i,j}].$$

Therefore, it holds that $V_{\theta,i,j}$ is zero-mean and $\nu = 1$ sub-Gaussian under the conditional distribution $\mathbb{P}_\theta[\cdot|x_{i,j}]$.

Now the Hessian of the loss function is given by

$$\nabla^2 \hat{l}_{D,\varepsilon}(\theta) = \frac{1}{nm} \sum_{i=1}^n \sum_{j=1}^m \left[ 1(\tilde{y}_{i,j} = 1) \left( (1 - \sigma(\varepsilon/m))\nabla^2 \log(1 - \sigma(\theta^\top x_{i,j})) - \sigma(\varepsilon/m)\nabla^2 \log \sigma(\theta^\top x_{i,j}) \right) \right.$$

$$\left. + 1(\tilde{y}_{i,j} = 0) \left( (1 - \sigma(\varepsilon/m))\nabla^2 \log \sigma(\theta^\top x_{i,j}) - \sigma(\varepsilon/m)\nabla^2 \log(1 - \sigma(\theta^\top x_{i,j})) \right) \right],$$

where

$$\nabla^2 \log \sigma(\theta^\top x_{i,j}) = \frac{\sigma''(\theta^\top x_{i,j})\sigma(\theta^\top x_{i,j}) - \sigma'(\theta^\top x_{i,j})^2}{\sigma(\theta^\top x_{i,j})^2} x_{i,j} x_{i,j}^\top,$$

$$\nabla^2 \log(1 - \sigma(\theta^\top x_{i,j})) = -\frac{\sigma''(\theta^\top x_{i,j})(1 - \sigma(\theta^\top x_{i,j})) + \sigma'(\theta^\top x_{i,j})^2}{(1 - \sigma(\theta^\top x_{i,j}))^2} x_{i,j} x_{i,j}^\top.$$

Now the second derivative of the logistic function $\sigma(\cdot)$ is given by $\sigma''(z) = \sigma'(z)(1 - 2\sigma(z))$, which gives us

$$\nabla^2 \log \sigma(\theta^\top x_{i,j}) = \nabla^2 \log(1 - \sigma(\theta^\top x_{i,j})) = -\sigma'(\theta^\top x_{i,j}) x_{i,j} x_{i,j}^\top.$$

Hence, the Hessian of the loss function takes the form

$$\nabla^2 \hat{l}_{D,\varepsilon}(\theta) = \frac{1}{nm} \sum_{i=1}^n \sum_{j=1}^m \left[ 1(\tilde{y}_{i,j} = 1)(2\sigma(\varepsilon/m) - 1)\sigma'(\theta^\top x_{i,j}) + 1(\tilde{y}_{i,j} = 0)(2\sigma(\varepsilon/m) - 1)\sigma'(\theta^\top x_{i,j}) \right] x_{i,j} x_{i,j}^\top.$$

Now, under Assumption 1, observe that $\sigma'(\theta^\top x_i) \geq \gamma$ for all $\theta \in \Theta_B$, where $\gamma = \frac{1}{2 + \exp(-2LB) + \exp(2LB)}$. This implies that $\hat{l}_{D,\varepsilon}$ is $\gamma_\varepsilon := \gamma(2\sigma(\varepsilon/m) - 1)$ strongly convex in $\Theta_B$ for all $\varepsilon > 0$ with respect to the semi-norm . Since $\theta^* \in \Theta_B$, introducing the error vector $\Delta = \hat{\theta}_{RR} - \theta^*$, we conclude that

$$\gamma_\varepsilon \|\Delta\|_{\Sigma_D}^2 \leq \left\| \nabla \hat{l}_{D,\varepsilon}(\theta^*) \right\|_{\Sigma_D^{-1}}^2 \|\Delta\|_{\Sigma_D}$$

Introducing $M = \frac{1}{n^2} X\Sigma_D^{-1}X^\top$, we now have $\|\nabla l_{D,\varepsilon}(\theta^*)\|_{\Sigma_D^{-1}}^2 = V_{\theta^*}^\top M V_{\theta^*}$. Then, the Bernstein's inequality for sub-Gaussian random variables in quadratic form (see e.g. Theorem 2.1 in Hsu et al. (2012)) implies that with probability at least $1 - \alpha$,

$$\left\| \nabla \hat{l}_{D,\varepsilon}(\theta^*) \right\|_{\Sigma_D^{-1}}^2 = V_{\theta^*}^\top M V_{\theta^*} \leq \nu^2 \left( \operatorname{tr}(M) + 2\sqrt{\operatorname{tr}(M^\top M)\log(1/\alpha)} + 2\|M\|\log(1/\alpha) \right)$$

$$\leq C_1 \cdot \nu^2 \cdot \frac{d + \log(1/\alpha)}{nm}$$

For some $C_1 > 0$. This gives us

$$\gamma_\varepsilon \|\Delta\|_{\Sigma_D+} \leq \left\| \nabla \hat{l}_{D,\varepsilon}(\theta^*) \right\|_{\Sigma_D^{-1}} \leq \sqrt{C_1 \cdot \nu^2 \cdot \frac{d + \log(1/\alpha)}{nm}}.$$

Solving for the above inequality, we get

$$\|\Delta\|_{\Sigma_D} \leq C_2 \cdot \sqrt{\frac{\nu^2}{\gamma_\varepsilon^2} \cdot \frac{d + \log(1/\alpha)}{nm}}$$

for some constant $C_2 > 0$. Now note that $\frac{\nu}{\gamma_\varepsilon} = \frac{1}{\gamma} \cdot \frac{e^{\varepsilon/m} + 1}{e^{\varepsilon/m} - 1}$. Hence, we get

$$\left\| \hat{\theta}_{RR} - \theta^* \right\|_{\Sigma_D} \leq \frac{C}{\gamma} \cdot \frac{e^{\varepsilon/m} + 1}{e^{\varepsilon/m} - 1} \sqrt{\frac{d + \log(1/\alpha)}{nm}},$$

Equivalently,

$$\|\hat{\theta}_{RR} - \theta^*\| \leqslant \frac{C}{\gamma\sqrt{\lambda_{\min}(\Sigma_{\mathcal{D}})}} \cdot \frac{e^{\varepsilon/m}+1}{e^{\varepsilon/m}-1} \sqrt{\frac{d + \log(1/\alpha)}{nm}},$$

for some $C > 0$, which holds for any $\varepsilon \in (0, \infty)$. This completes our proof. $\square$

## C.2  Proof of Theorem 3

*Proof.* As the subsets $D_i$ are disjoint. It is sufficient to show Algorithm 2 is $(\varepsilon, \delta)$ user-level (label) DP.

To prove the privacy guarantee, note that there are two components related to privacy in each iteration of Algorithm 2, the AboveThreshold (step 9) and private mean oracle (step 10-15). For the AboveThreshold, as the query $s_t^c$. Thus, by Dwork et al. (2014), we can see it will be $\frac{\varepsilon}{2}$-DP for $T$ iterations in total. Thus, it is sufficient for us to show steps 10-15 are $(\frac{\varepsilon}{2}, \delta)$-user-level (label) DP.

Since it consists of $T$-folds, we first recall the advanced composition theorem Dwork et al. (2014).

**Lemma 1** (Advanced Composition Theorem Dwork et al. (2014))**.** *Given target privacy parameters $0 < \varepsilon < 1$ and $0 < \delta < 1$, to ensure $(\varepsilon, T\delta' + \delta)$-DP over T mechanisms, it suffices that each mechanism is $(\varepsilon', \delta')$-DP, where $\varepsilon' = \frac{\varepsilon}{2\sqrt{2T\ln(2/\delta)}}$ and $\delta' = \frac{\delta}{T}$.*

Thus, we have to show each iteration satisfies $(\varepsilon', \delta')$-DP, where $\varepsilon' = \frac{\varepsilon}{4\sqrt{2T\ln(2/\delta)}}$ and $\delta' = \frac{\delta}{T}$.
We then recall the following privacy amplification via Poison subsampling. $\square$

**Lemma 2** ((Bassily et al., 2014; Beimel et al., 2010))**.** *Let $A$ be an $(\varepsilon, \delta)$-DP algorithm. Now we construct the algorithm $B$ as follows: On input $D = \{x_1, \cdots, x_n\}$, first we construct a new sub-sampled dataset $D_S$ where each $x_i \in D_s$ with probability $q$. Then we run algorithm $A$ on the dataset $D_S$. Then $B(D) = A(D_S)$ is $(\tilde{\varepsilon}, \tilde{\delta})$-DP, where $\tilde{\varepsilon} = O(q\varepsilon)$ and $\tilde{\delta} = q\delta$.*

Note that when there is no subsampling, i.e., the batch size is $n$. Then when $\varsigma = O(\frac{\varepsilon}{\sqrt{T\ln(1/\delta)}})$, it will be the private mean oracle as in Liu & Asi (2024b). Thus, it will be $(\varepsilon', \delta')$-user-level (label) DP and the $T$-fold composition is $(\varepsilon, \delta)$-user-level (label) DP.

Thus, when there is Poison subsampling with the probability $\frac{\tilde{n}}{n}$, the whole algorithm (step 10-15) will be $(O(\frac{\tilde{n}}{n}\varepsilon), \frac{\tilde{n}}{n}\delta)$-user-level (label) DP. That is when we take $\varsigma = \tilde{O}(\frac{n}{\tilde{n}}\frac{\varepsilon}{\sqrt{T\ln(1/\delta)}})$, it will be $(\varepsilon', \delta')$-user-level (label)DP and the $T$-fold composition is $(\varepsilon, \delta)$-user-level (label) DP.

## C.3  Proof of Theorem 4

We begin with the definition of Lipschitz continues and strongly convexity.

**Definition 4. (Lipschitz Continuity)** A function $f : \mathbb{R}^n \to \mathbb{R}$ is said to be Lipschitz continuous with constant $L$ if there exists a constant $L \geq 0$ such that for all $x, y \in \mathbb{R}^n$,

$$\|f(x) - f(y)\| \leq L\|x - y\|$$

where $\|\cdot\|$ denotes the norm in $\mathbb{R}^n$ (typically the Euclidean norm).

**Definition 5. (Strong Convexity)** A function $f : \mathbb{R}^n \to \mathbb{R}$ is said to be strongly convex with parameter $\mu > 0$ if there exists a constant $\mu > 0$ such that for all $x, y \in \mathbb{R}^n$, we have:

$$f(y) \geq f(x) + \nabla f(x)^\top (y - x) + \frac{\mu}{2}\|y - x\|^2$$

where $\mu$ is the strong convexity parameter.

We can see when we use the full batch in Algorithm 2, i.e., $\tilde{n}_i = n_i$, based on our parameters, Algorithm 1 will be the same as the method in Liu & Asi (2024b) for DP-SCO with strongly convex loss in the user-level DP setting. Specifically, we have the following result by Theorem 4.11 in Liu & Asi (2024b)

**Lemma 3** (Derived by Theorem 4.11 in Liu & Asi (2024b)). *For $0 < \varepsilon < 10, 0 < \delta < 1$, if the population loss function*

$$l_{\mathcal{P},\varepsilon}(\theta) = \mathbb{E}_{s \sim \rho(\cdot), (a^0, a^1) \sim \mu(\cdot|s)} \mathbb{I}\{y = 1\} \log \sigma(\theta^T x) + \mathbb{I}\{y = 0\} \log(1 - \sigma(\theta^T x))$$

*is $\kappa\gamma$-strongly convex and $4L$-Lipschitz continuous, under the same parameters in Theorem 4, Algorithm 1 outputs $\widehat{\theta}$ such that*

$$\mathbb{E}\left[L_P(\widehat{\theta}) - \min_{\theta^\star \in \Theta} L_P(\theta^\star)\right] \leq O\left(\frac{L^2}{\kappa\gamma} \cdot \left(\frac{1}{nm} + \frac{d \log^2(ndm/\delta)}{n^2 m \varepsilon^2}\right)\right)$$

Thus, it is sufficient for us to show that population loss function is $\kappa\gamma$-strongly convex and $4L$-Lipschitz continuous under Assumption 1 and 2.

**Proof of Theorem 4** By strongly convexity, we have

$$\mathbb{E}\|\hat{\theta}_k - \theta^*\|^2 \leq \frac{2}{\kappa\gamma}\mathbb{E}\left[\widehat{l}_{\mathcal{P},\varepsilon}(\hat{\theta}_k) - \widehat{l}_{\mathcal{P},\varepsilon}(\theta^*)\right]$$

$$\leq O\left(\frac{L^2}{\kappa^2\gamma^2} \cdot \left(\frac{1}{nm} + \frac{d \log^2(ndm/\delta)}{n^2 m \varepsilon^2}\right)\right)$$

The second inequality comes from Lemma 3.

**Lemma 4.** *Under Assumption 1 and 2, the loss function*

$$l_{\mathcal{P},\varepsilon}(\theta) = \mathbb{E}_{s \sim \rho(\cdot), (a^0, a^1) \sim \mu(\cdot|s)} \mathbb{I}\{y = 1\} \log \sigma(\theta^T x) + \mathbb{I}\{y = 0\} \log(1 - \sigma(\theta^T x))$$

*is $\kappa\gamma$-strongly convex and $4L$-Lipschitz continuous.*

*Proof of Lemma 4.* **Lipschitz continuous**

The gradient of the loss function is given by

$$\nabla l_{\mathcal{P},\varepsilon}(\theta) = \mathbb{E}_{s \sim \rho(\cdot), (a^0, a^1) \sim \mu(\cdot|s)} [V_\theta x]$$

where

$$V_\theta = \mathbb{I}\{y = 1\}\frac{\sigma'(\theta^T x)}{\sigma(\theta^T x)} - \mathbb{I}\{y = 0\}\frac{\sigma'(\theta^T x)}{1 - \sigma(\theta^T x)}.$$

Furthermore, we have

$$|V_\theta|_{y=1} = \frac{\sigma'(\theta^T x)}{\sigma(\theta^T x)} = 1 - \sigma(\theta^T x) \leq 1.$$

$$|V_\theta|_{y=0} \leq 1.$$

Therefore, by Assumption 1, it holds that

$$|\nabla l_{\mathcal{P},\varepsilon}(\theta)| \leq |V_\theta| \cdot |x| \leq 4L$$

This means $l_{\mathcal{P},\varepsilon}(\theta)$ is $4L$ - Lipschitz continuous
**Strongly convex**
Now, the Hessian of the loss function is given by

$$\nabla^2 l_{\mathcal{P},\varepsilon}(\theta) = \mathbb{E}_{s \sim \rho(\cdot), (a^0, a^1) \sim \mu(\cdot|s)} \left[\mathbb{I}\{y = 1\}\sigma'(\theta^T x) + \mathbb{I}\{y = 0\}\sigma'(\theta^T x)\right] xx^T.$$

we observe that $\sigma'(\theta^T x) \geq \gamma$ for all $\theta \in \Theta_B$, where

$$\gamma = \frac{1}{2 + \exp(-2LB) + \exp(2LB)}.$$

by Assumption 2,

$$v^T \nabla^2 l_{\mathcal{P},\varepsilon}(\theta) v \geq \kappa\gamma \|v\|_2^2.$$

This implies that $l_{\mathcal{P},\varepsilon}$ is $\kappa\gamma$-strongly convex in $\Theta_B$.

$\square$

### C.4 Proof of Theorem 5

We follow the proof of Theorem 4. First, we demonstrate that the objective loss function $\mathcal{L}_{\mathcal{P}}(\theta, \pi)$ is $\kappa\gamma$-strongly convex and $2LK^2$-Lipschitz continuous. Then, with the help of the lemma 3, the proof is completed.

**Lemma 5.** *We define $\mathcal{L}_{\mathcal{P}}(\theta, \pi) = \mathbb{E}_{s \sim \rho(\cdot), (a_1,\dots,a_K) \sim \mu(\cdot|s)} \ell(\theta; s, \pi)$. Under Assumption 1 and 2 , $\mathcal{L}_{\mathcal{P}}(\theta, \pi)$ is $\kappa\gamma$-strongly convex and $2LK^2$- Lipschitz continuous, where $\gamma = \exp(-4LB)/2$.*

***Proof of Lemma 5.*** Let $s$ be a state and $a_1, \dots, a_K$ be $K$ actions to be compared at that state. Let the label/preference feedback $y \in \{1, 2, \dots, K\}$ indicate which action is most preferred by the human labeler. Let

$$x_{i,j} = \phi(s, a_i) - \phi(s, a_j), \quad 1 \leq i \neq j \leq K$$

be the feature difference between actions $a_i$ and $a_j$ at state $s$. Define the population covariance matrix

$$\Sigma_{i,j} = \mathbb{E}_{s \sim \rho(\cdot), (a_1,\dots,a_K) \sim \mu(\cdot|s)} \left[ x_{i,j} x_{i,j}^\top \right].$$

**Assumption 3.** (Coverage of feature space) The data distributions $\rho$, $\mu$ are such that

$$\lambda_{\min}\left(\Sigma_{i,j}\right) \geq \kappa \quad \text{for some constant} \quad \kappa > 0 \quad \text{for all} \quad 1 \leq i \neq j \leq K.$$

$$\ell(\theta) = -\frac{1}{n} \sum_{i=1}^{n} \sum_{j=1}^{K} \log\left( \frac{\exp(\langle \theta, \phi(s^i, a^i_{\pi(j)}) \rangle)}{\sum_{k=j}^{K} \exp(\langle \theta, \phi(s^i, a^i_{\pi(k)}) \rangle)} \right)$$

**Strongly convexity of $\mathcal{L}_{\mathcal{P}}(\theta, \pi)$**(follow Proof of Theorem 4.1 in Zhu et al. (2023))

The Hessian of the negative log likelihood can be written as

$$\nabla^2 \mathcal{L}_P(\theta, \pi) = \mathbb{E} \sum_{j=1}^{K} \sum_{k=j}^{K} \frac{\exp\left(\langle \theta, \phi(s, a_{\pi(j)}) + \phi(s, a_{\pi(k)}) \rangle\right)}{\left(\sum_{k'=j}^{K} \exp\left(\langle \theta, \phi(s, a_{\pi(k')}) \rangle\right)\right)^2} \cdot \left(\phi(s, a_{\pi(j)}) - \phi(s, a_{\pi(k)})\right) \left(\phi(s, a_{\pi(j)}) - \phi(s, a_{\pi(k)})\right)^\top$$

Since $\exp(\theta, \phi) \in [\exp(-LB), \exp(LB)]$, we know that the coefficients satisfy

$$\frac{\exp\left(\langle \theta, \phi(s, a_{\pi(j)}) + \phi(s, a_{\pi(k)}) \rangle\right)}{\left(\sum_{k'=j}^{K} \exp\left(\langle \theta, \phi(s, a_{\pi(k')}) \rangle\right)\right)^2} \geq \frac{\exp(-4LB)}{2(K + 1 - j)^2}$$

Set $\gamma = \exp(-4LB)/2$. We can verify that for any vector $v \in \mathbb{R}^K$, one has

$$v^\top \nabla^2 \mathcal{L}_P(\theta, \pi) v \geq \mathbb{E}\gamma v^\top \left[ \sum_{j=1}^{K} \frac{1}{(K+1-j)^2} \sum_{k'=k}^{K} \sum_{k=j}^{K} \left(\phi(s, a_{\pi(j)}) - \phi(s, a_\pi)\right) \left(\phi(s, a_{\pi(j)}) - \phi(s, a_{\pi(k)})\right)^\top \right] v$$

$$\geq \mathbb{E}\gamma v^\top \left[ \min_{\pi \in \Pi(K)} \sum_{j=1}^{K} \frac{1}{(K+1-j)^2} \sum_{k=j}^{K} \sum_{k'=k}^{K} \left( \phi(s, a_{\pi(j)}) - \phi(s, a_{\pi(k)}) \right) \left( \phi(s, a_{\pi(j)}) - \phi(s, a_{\pi(k)}) \right)^\top \right] v$$

$$\geq \kappa\gamma \|v\|_2^2$$

The last inequality uses Assumption 3.
So, $\mathcal{L}_\mathcal{P}(\theta, \pi)$ is $\kappa\gamma$-strongly convex with respect to $\ell_2$-norm.

**Lipschitz continuous**
The gradient of the negative log likelihood is

$$\nabla \ell(\theta, \pi) = -\sum_{j=1}^{K} \sum_{k=j}^{K} \frac{\exp\left( \left\langle \theta, \phi\left(s, a_{\pi(k)}\right) \right\rangle \right)}{\sum_{k'=j}^{K-1} \exp\left( \left\langle \theta, \phi\left(s, a_{\pi(k')}\right) \right\rangle \right)} \cdot \left( \phi\left(s, a_{\pi(j)}\right) - \phi\left(s, a_{\pi(k)}\right) \right).$$

We set $x_{jk} = \phi(s, a_j) - \phi(s, a_k)$. $X \in \mathbb{R}^{(K(K-1)/2) \times d}$ has the differencing vector $x_{jk}$ as its $(k + \sum_{l=K-j+1}^{K} l)^{\text{th}}$ row. We also define $V_{jk}$ to be the random variable of the coefficient of $x_{jk}$ in equation above under the PL model, i.e., conditioned on an arbitrary permutation $\pi$.

$$V_{jk} = \begin{cases} \frac{\exp(\langle \theta, \phi(s, a_k) \rangle)}{\sum_{k'=\pi^{-1}(j)}^{K} \exp\left( \left\langle \theta, \phi\left(s, a_{\pi(k')}\right) \right\rangle \right)}, & \text{if } \pi^{-1}(j) < \pi^{-1}(k) \\ -\frac{\exp(\langle \theta, \phi(s, a_j) \rangle)}{\sum_{k'=\pi^{-1}(k)}^{K} \exp\left( \left\langle \theta, \phi\left(s, a_{\pi(k')}\right) \right\rangle \right)}, & \text{otherwise.} \end{cases}$$

Here, $\pi^{-1}(j) < \pi^{-1}(k)$ means that the $j$-th item ranks higher than the $k$-th item. Let $V \in \mathbb{R}^{K(K-1)/2}$ be the concatenated random vector of $\{V_{jk}\}_{1 \leq j < k \leq K}$. Furthermore, since under any permutation, the sum of the absolute value of each element in $V$ is at most $K$. Thus,

$$\|\nabla \ell(\theta, \pi)\|_2^2 = V^\top X X^\top V \leq \|X\|_2^2 \|V\|_2^2 \leq 4L^2 K^4$$

Thus, $\nabla \ell(\theta, \pi)$ is $2K^2 L$-Lipschitz continuous, and $\mathcal{L}_P(\theta, \pi)$ is also $2K^2 L$-Lipschitz continuous. $\qquad \square$

The rest of the proof is the same as that of Theorem 4.

### C.5 Proof of Theorem 6

To establish our lower bounds, we begin by introducing key background concepts, notations, and properties. Consider a family of distributions $\mathcal{P}$ over $(\mathcal{X}^m)^n$, where $\mathcal{X}$ represents the data universe, $m$ denotes the sample size, and $n$ refers to the user size. Our goal is to estimate a parameter $\theta$, which is a mapping $\theta : \mathcal{P} \to \theta$ that characterizes the underlying distribution.

To quantify the estimation error, we define a pseudo-metric $\rho : \Theta \times \Theta \to \mathbb{R}_+$, which serves as the loss function for evaluating the accuracy of the estimate. The minimax risk under the loss function $\rho$ for the class $\mathcal{P}$ is given by:

$$R(\mathcal{P}, \rho) := \min_{\hat{\theta}} \max_{P \in \mathcal{P}} \mathbb{E}_{X \sim P}[\rho(\hat{\theta}(X), \theta(P))].$$

In this study, we focus on estimating an unknown parameter, $\theta$, within the Bradley-Terry-Luce model under a use-level privacy framework. Data is collected from $n$ users, where each user provides $m$ samples. We consider a fixed design setup, meaning that the feature

vectors $x_{i,j} \in \mathbb{R}^d$ for samples $i \in [n]$ from user $j \in [m]$ are predetermined and known. Our objective is to infer $\theta$ based on a sequence of private responses $\tilde{y}_{i,j}$.

Without privacy constraints, responses $y_{i,j}$ follow a logistic model:

$$\mathbb{P}\left(y_{i,j} = 1 \mid x_{i,j}\right) = \sigma\left(\theta^\top x_{i,j}\right) = \frac{1}{1 + \exp\left(-\theta^\top x_{i,j}\right)}, \quad \mathbb{P}\left(y_{i,j} = 0 \mid x_{i,j}\right) = 1 - \sigma\left(\theta^\top x_{i,j}\right). \tag{4}$$

We refer to this distribution family as $\mathcal{P}_\theta$. Under the use-level privacy constraint, our goal is to construct an estimator $\hat{\theta}$ that closely approximates $\theta$ while preserving use-level-label differential privacy, ensuring that an entire user's labels remain protected. Depending on the estimation framework, the loss function $\rho$ is defined as either the squared $\ell_2$-norm or a squared semi-norm.

**Lemma 6** (Assouad's lemma Chowdhury et al. (2024b)). *Let $\mathcal{V} \subseteq (\mathcal{P})^m$ be a set of distributions indexed by the hypercube $\mathcal{E}_d = \{\pm 1\}^d$. Suppose there exists a $\tau \in \mathbb{R}$ and $\alpha > 0$, such that $\rho$ satisfies: (i) for all $u, v, w \in \mathcal{E}_d$, $\rho\left(\theta\left(P_u\right), \theta\left(P_v\right)\right) \geq 2\tau \cdot \sum_{i=1}^d \mathbb{1}\left(u_i \neq v_i\right)$ and (ii) $\rho\left(\theta\left(P_u\right), \theta\left(P_v\right)\right) \leq \alpha\left(\rho\left(\theta\left(P_u\right), \theta\left(P_w\right)\right) + \rho\left(\theta\left(P_v\right), \theta\left(P_w\right)\right)\right)$, i.e., $\alpha$-triangle inequality. For each $i \in [d]$, define the mixture distributions:*

$$P_{+i} := \frac{2}{|\mathcal{E}_d|} \sum_{e \in \mathcal{E}_d : e_i = 1} P_e \text{ and } P_{+i} := \frac{2}{|\mathcal{E}_d|} \sum_{e \in \mathcal{E}_d : e_i = -1} P_e.$$

*Then, we have*

$$R(\mathcal{P}, \rho) \geq \frac{\tau}{2\alpha} \sum_{i=1}^d \left(1 - \|P_{+i} - P_{-i}\|_{\mathrm{TV}}\right).$$

**Corollary 2.** *Under the same conditions of Lemma 6, we have*

$$R(\mathcal{P}, \rho) \geq \frac{d\tau}{2\alpha} \left[1 - \left(\frac{1}{d} \sum_{i=1}^d \frac{1}{2^d} \sum_{e \in \mathcal{E}_d} \|P_e - P_{\bar{e}^i}\|_{\mathrm{TV}}^2\right)^{1/2}\right].$$

*where $\bar{e}^i$ is a vector in $\mathcal{E}_d$ that flips the $i$-th coordinate of $e$.*

**Lemma 7** (Assouad's lemma Chowdhury et al. (2024b)). *Let the same conditions of Lemma 6 hold. If for all $i \in [d]$, there exists a coupling $(X, Y)$ between $P_{+i}$ and $P_{-i}$ with $\mathbb{E}\left[d_{\mathrm{Ham}}(X, Y)\right] \leq D$ for some $D \geq 0$, then*

$$R(\mathcal{P}, \rho, \varepsilon, \delta) \geq \frac{d\tau}{2\alpha} \cdot \left(0.9 e^{-10\varepsilon D} - 10D\delta\right).$$

**Fact 1.** *Let $p_a = \frac{1}{1+e^i}$ and $p_b = \frac{1}{1+e^b}$. Then, the sum of KL-divergences between the corresponding Bernoulli distributions satisfies:*

$$\mathrm{kl}\left(p_a \| p_b\right) + \mathrm{kl}\left(p_b \| p_a\right) \leq (a - b)^2,$$

*where $\mathrm{kl}(p\|q)$ represents the KL-divergence between Bernoulli distributions with parameters $p$ and $q$ $D_{\mathrm{KL}}(\mathrm{Bernoulli}(p)\|\mathrm{Bernoulli}(q))$.*

**Proof of Fact 1.** By direct calculation, the sum of KL-divergences simplifies to:

$$\mathrm{kl}\left(p_a \| p_b\right) + \mathrm{kl}\left(p_b \| p_a\right) = (p_a - p_b) \log\left(\frac{p_a}{1 - p_a} \cdot \frac{1 - p_b}{p_b}\right).$$

From the definitions of $p_a$ and $p_b$, we substitute:

$$(p_a - p_b) \log\left(\frac{p_a}{1 - p_a} \cdot \frac{1 - p_b}{p_b}\right) = \left(\frac{1}{1 + e^a} - \frac{1}{1 + e^b}\right) \cdot (b - a).$$

Now, assuming without loss of generality that $b \geq a$, we bound the difference:

$$\frac{1}{1 + e^a} - \frac{1}{1 + e^b} \leq \frac{e^b - e^a}{e^b} = 1 - e^{a-b}.$$

Using the inequality $e^{a-b} \geq 1 + (a-b)$, we obtain:

$$1 - e^{a-b} \leq b - a.$$

Thus, combining these results gives the final bound:

$$\mathrm{kl}\left(p_a \| p_b\right) + \mathrm{kl}\left(p_b \| p_a\right) \leq (a-b)^2.$$

This completes the proof. $\qquad\square$

Now, we proceed to prove Theorem 6, which we break down into two parts: the non-private component and the private component.

**Non-private component.** We begin by selecting a parameter $\Delta > 0$ and defining $\theta_e = \Delta e$ for each $e \in \mathcal{E}_d = \{\pm 1\}^d$. Each $P_e$ is a probability distribution associated with a specific vector $e \in \mathcal{E}_d$. This means that for every binary vector $e = (e_1, e_2, \ldots, e_d)$, there is a corresponding probability distribution $P_e$. Since we consider each user provides $m$ samples, thus here $\mathcal{P}_\theta^{\otimes m}$ is a $m$-product distribution. Our goal is to verify the two conditions stated in Lemma 6.

First, we observe that the function $\rho = \|\cdot\|_2^2$ adheres to the 2-triangle inequality, meaning $\alpha = 2$. Additionally, for any $u, v \in \mathcal{E}_d$, the squared norm difference satisfies

$$\|\theta_u - \theta_v\|_2^2 = 4\Delta^2 \sum_{i=1}^d \mathbb{1}\left(u_i \neq v_i\right),$$

indicating that $\tau = 2\Delta^2$. Now, let $P_e^n$ represent the distribution corresponding to $n$ independent samples from $m$ users of the (non-private) observations $y_{i,j}$ when $\theta = \theta_e$. Then, by applying Corollary 2, we obtain

$$R\left(\mathcal{P}_\theta^{\otimes m}, \|\cdot\|_2^2, \varepsilon, \delta\right) \geq R\left(\mathcal{P}_\theta^{\otimes m}, \|\cdot\|_2^2\right) \geq \frac{d\Delta^2}{2}\left[1 - \left(\frac{1}{d}\sum_{i=1}^d \frac{1}{2^d}\sum_{e\in\mathcal{E}_d}\left\|P_e^n - P_{\bar{e}^i}^n\right\|_{\mathrm{TV}}^2\right)^{1/2}\right].$$

Using Pinsker's inequality along with the chain rule for KL-divergence, we obtain the following bound by bounding the total variation (TV) distance for any $u, v \in \mathcal{E}_d$:

$$\left\|P_u^n - P_v^n\right\|_{\mathrm{TV}}^2 \leq \frac{1}{4}\left(D_{\mathrm{KL}}\left(P_u^n \| P_v^n\right) + D_{\mathrm{KL}}\left(P_v^n \| P_u^n\right)\right)$$

$$= \frac{1}{4}\sum_{j=1}^m \sum_{k=1}^n \left(\mathrm{kl}\left(p_u\left(x_{k,j}\right) \| p_v\left(x_{k,j}\right)\right) + \mathrm{kl}\left(p_v\left(x_{k,j}\right) \| p_u\left(x_{k,j}\right)\right)\right).$$

Next, applying Fact 1, we derive an upper bound on the total variation distance:

$$\left\|P_u^n - P_v^n\right\|_{\mathrm{TV}}^2 \leq \frac{\Delta^2}{4}\sum_{j=1}^m \sum_{k=1}^n \left(x_{k,j}^\top(u-v)\right)^2.$$

Moreover, we have

$$\frac{1}{d2^d}\sum_{i=1}^d \sum_{e\in\mathcal{E}_d}\left\|P_e^n - P_{\bar{e}^i}^n\right\|_{\mathrm{TV}}^2 \leq \frac{\Delta^2}{4d}\frac{1}{2^d}\sum_{e\in\mathcal{E}_d}\sum_{i=1}^d \sum_{j=1}^m \sum_{k=1}^n \left(2x_{k,j,i}\right)^2.$$

Rewriting in terms of the Frobenius norm,

$$\frac{1}{d2^d}\sum_{i=1}^d \sum_{e\in\mathcal{E}_d}\left\|P_e^n - P_{\bar{e}^i}^n\right\|_{\mathrm{TV}}^2 = \frac{\Delta^2}{d}\frac{1}{2^d}\sum_{e\in\mathcal{E}_d}\sum_{i=1}^d \sum_{j=1}^m \sum_{k=1}^n x_{k,j,i}^2 = \frac{\Delta^2}{d}\frac{1}{2^d}\sum_{e\in\mathcal{E}_d}\|X\|_F^2.$$

Here, $X \in \mathbb{R}^{mn\times d}$ represents the aggregated data matrix, where each row $x_{k,j}^\top \in \mathbb{R}^d$ corresponds to a sample from user $j$, and $\|\cdot\|_F$ denotes the Frobenius norm. Using this bound, we derive the following lower bound:

$$R\left(\mathcal{P}_\theta^{\otimes m}, \|\cdot\|_2^2, \varepsilon, \delta\right) \geq \frac{d\Delta^2}{2}\left[1 - \left(\frac{\Delta^2}{d}\|X\|_F^2\right)^{1/2}\right].$$

To optimize the bound, we set:

$$\Delta^2 = \frac{d}{4\|X\|_F^2}.$$

which simplifies the expression to:

$$R\left(\mathcal{P}_\theta^{\otimes m}, \|\cdot\|_2^2, \varepsilon, \delta\right) \geq \frac{d^2}{16\|X\|_F^2} = \frac{d}{mn} \cdot \frac{1}{16\frac{1}{dmn}\sum_{j=1}^m \sum_{k=1}^n \left\|x_{k,j}\right\|^2}.$$

Given the assumption that $\left\|x_{k,j}\right\|^2 \leq L^2$, we further refine the bound as:

$$R\left(\mathcal{P}_\theta^{\otimes m}, \|\cdot\|_2^2, \varepsilon, \delta\right) \geq \Omega\left(\frac{d}{L^2} \cdot \frac{d}{mn}\right).$$

Now we consider the **Private component**. Similar to the non-private part, we verify $\Delta, \theta_e, \rho$. Since we consider user-level privacy, we will assume $\mathcal{P}_\theta^{\otimes m}$ is an m-product distribution. Applying Lemma 6, we obtain the following lower bound:

$$R\left(\mathcal{P}_\theta^{\otimes m}, \|\cdot\|_2^2, \varepsilon, \delta\right) \geq \frac{d\Delta^2}{2}\left(0.9e^{-10\varepsilon D} - 10D\delta\right).$$

By using the fact that $e^z \geq 1 + x$, we further derive:

$$R\left(\mathcal{P}_\theta^{\otimes m}, \|\cdot\|_2^2, \varepsilon, \delta\right) \geq \frac{d\Delta^2}{2}(0.9 - 10D(\varepsilon + \delta)).$$

Here, $D$ represents an upper bound on the expected Hamming distance between $(X, Y)$, where $(X, Y)$ is a coupling of $P_{+i}^n$ and $P_{-i}^n$. Our next step is to determine an appropriate bound for $D$, which requires analyzing the expected Hamming distance between these two product distributions. For a lower bound, it suffices to consider $x_{k,j} = x \in \mathbb{R}^d$ where $\|x\|_\infty \leq 1$ for all $k \in [n], j \in [m]$. Applying the standard result on maximal coupling, we obtain:

$$\mathbb{E}\left[d_{\text{Ham}}(X, Y)\right] = n\left\|P_{+i} - P_{-i}\right\|_{\text{TV}}.$$

Using above and the joint convexity of TV distance, we derive:

$$\|P_{+i} - P_{-i}\|_{\text{TV}} = \left\|\frac{1}{|\mathcal{E}_d|}\sum_{e \in \mathcal{E}_e} P_{e,+i} - P_{e,-i}\right\|_{\text{TV}}$$

$$\leq \frac{1}{|\mathcal{E}_d|}\sum_{e \in \mathcal{E}_d}\|P_{e,+i} - P_{e,-i}\|_{\text{TV}}$$

$$\leq \max_{e \in \mathcal{E}_d, i \in [d]}\|P_{e,+i} - P_{e,-i}\|_{\text{TV}}$$

$$= \max_{e \in \mathcal{E}_b, i \in [d]}\|P_e - P_{\bar{e}^i}\|_{\text{TV}}.$$

Here, $\bar{e}^i$ represents a modified version of $e$ where its $i$-th coordinate is flipped. Applying Pinsker's inequality for any $i \in [d]$, we obtain:

$$\|P_e - P_{\bar{e}^i}\|_{\text{TV}}^2 \leq \frac{1}{4}\left(D_{\text{KL}}\left(P_e\|P_{\bar{e}^i}\right) + D_{\text{KL}}\left(P_e\|P_{\bar{e}^i}\right)\right) \leq m\Delta^2.$$

The last inequality follows from Fact 1, the assumption that $\|x\|_\infty \leq 1$ and the conclusion in Levy et al. (2021). Combining these results, we establish:

$$\mathbb{E}\left[d_{\text{Ham}}(X, Y)\right] = n\left\|P_{+i} - P_{-i}\right\|_{\text{TV}} \leq \sqrt{m}n\Delta := D.$$

With the derived value of $D$, we now arrive at the following lower bound:

$$R\left(\mathcal{P}_\theta^{\otimes m}, \|\cdot\|_2^2, \varepsilon, \delta\right) \geq \frac{d\Delta^2}{2}(0.9 - 10\sqrt{m}n\Delta(\varepsilon + \delta)).$$

To optimize this bound, we select

$$\Delta = \frac{0.04}{\sqrt{mn}(\varepsilon + \delta)}.$$

Substituting this choice into the expression, we obtain:

$$R\left(\mathcal{P}_\theta^{\otimes m}, \|\cdot\|_2^2, \varepsilon, \delta\right) \geq c \cdot \frac{d}{mn^2(\varepsilon + \delta)^2}.$$

for some universal constant $c$. Finally, combining this result with the non-private case completes the proof.

