# OpenReview forum: "Towards User-level Private Reinforcement Learning with Human Feedback"
_colmweb.org/COLM/2025/Conference — COLM 2025_

### Official Review · Reviewer_Kvf8 · 2025-05-09

**Rating:** 4
**Confidence:** 3
**Ethics Flag:** 1

**Summary:**

The paper proposes a new privacy-aware approach to RLHF (AUP-RLHF). The method satisfies user level differential privacy and is based on computing the average gradient of the loss with respect to each user’s preferences for updating model parameters during RL. Experiments include sentiment analysis and summarization tasks using two model families of different sizes (Gemma2-2b and Llama2-7b).

**Questions To Authors:**

* You propose AUP-RLHF but what AUP stands for is never defined in the abstract or introduction.
* You introduce mathematical expressions in the introduction but these are not clearly defined, i.e. what each variable represents?

**Reasons To Accept:**

* The proposed method looks novel and well motivated (however the overall narrative is hard to follow, see comments below).

**Reasons To Reject:**

* The experimental setting is not very clear. The authors should provide details on what are the private attributes in each dataset.*
* No results are reported for Llama in the summarization task. In general, the paper lacks consistency in the way that the results are presented and discussed.
* In general, the paper is really dense and can be improved by providing more clear/simple definitions of the underlying concepts presented.

---

> ### Author Response · Authors · 2025-06-02
>
> - **W1: Question about experimental setting**
> >the experimental setting is not very clear. The authors should provide details on what are the private attributes in each dataset.
>
> **Response to W1:**
>
> Thanks for the comment. We have already mentioned in the introduction that we consider the case where the user's preference is private, which follows the same setting as in the previous work. Also this is reasonable in practice.
>
> The IMDb and TL;DR datasets both follow the data format described in 125 rows, where each data point is given by $Z_i := ( s_{i,j}, a^0_{i,j}, a^1_{i,j}, y_{i,j} )_{j=1}^m,$
>
> in which $s_{i,j}$ is the prompt given to the LLM, $a^0_{i,j}$ and $a^1_{i,j}$ are two responses generated by the model, and $y_{i,j} \in \{0,1\}$ indicates the user's preference, whether $a^0_{i,j}$ is preferred over $a^1_{i,j}$. Among these, only $y_{i,j}$ is considered a private attribute, while $s_{i,j}$, $a^0_{i,j}$, and $a^1_{i,j}$ are public.
>
> ---
>
> - **W2: Add experimental results**
> >No results are reported for Llama in the summarization task. In general, the paper lacks consistency in the way that the results are presented and discussed.
>
> **Response to W2:**
>
> Thanks for the comment. We added experimental results on the TL;DR Summarization task using Llama-2 7B, under the same settings as in Tables 3, 4, and 5. The results show that our AUP-RLHF consistently achieves higher win rate than other baselines at both $ \varepsilon = 3$ and $\varepsilon = 8$, demonstrating the superior utility of our method.
>
> Table : Win Rate Against the SFT model for TL;DR Summarization (Llama-2-7b)
> | epsilon            | 3      | 8      |
> |--------------------|--------|--------|
> | AUP-RLHF           | 45%    | 58%    |
> | User-wise DPSGD    | 42%    | 55%    |
> | Random Response    | 41%    | 49%    |
> | Group Privacy      | 41%    | 52%    |
>
> ---
>
> - **W3: Presentation issue**
> >In general, the paper is really dense and can be improved by providing more clear/simple definitions of the underlying concepts presented.
>
> - **Q1: Presentation issue**
> >You propose AUP-RLHF but what AUP stands for is never defined in the abstract or introduction.
>
> - **Q2: Presentation issue**
> >You introduce mathematical expressions in the introduction but these are not clearly defined, i.e. what each variable represents?
>
> **Response to W3, Q1 and Q2:**
>
> Thanks for the comment. Due to space limitations, some explanations were brief and we will provide more clear/simpler definitions in Appendix/revised version as you suggested. More specifically, AUP stands for Adaptive User Private.
>
> And we list all the mathematical expressions mentioned in the Introduction along with their corresponding meanings in the table below. We will add these clarifications in our revised version.
> | Parameter | Meaning |
> |----------|---------|
> | $n$ | Number of users |
> | $m$ | Contributions per user |
> | $\theta$ | Reward model parameter |
> | $\theta^*$ | Ground-truth reward model parameter |
> | $d$ | Dimension of $\theta$ |
> | $\varepsilon$ | Privacy budget |
> | $\delta$ | Failure probability |
> | $K$ | Number of candidate responses |
> | $\sigma$ | Sigmoid function |
> | $r_\theta$ | Reward function parameterized by $\theta$ |
> | $\phi$ | Feature map |
> | $x$ | Differential feature vector |
> | $y$ | Human preference label |
> | $\tilde{y}$ | Randomized label |
> | $\Sigma$ | Covariance matrix |
> | $\lambda_{\min}$ | Minimum eigenvalue |
> | $\gamma$ | Strong convexity parameter |
> | $L$ | Lipschitz constant |
> | $B$ | Bound on $\|\theta\|$ |
> | $\kappa$ | Coverage parameter |
> | $\tau$ | Concentration threshold |
> | $C$ | Gradient clipping bound |
> | $\alpha$ | Probability bound |
> | $\eta$ | Learning rate |
> | $T$ | Number of iterations |
> | $\bar{n}$ | Batch size of users |
> | $\mathcal{D}$ | Dataset |
> | $\mathcal{P}$ | Distribution family |
> | $\rho$ | Pseudo-metric |
> | $\Delta$ | Parameter difference |
> | $D$ | Hamming distance bound |

---

> > ### Comment · Reviewer_Kvf8 · 2025-06-03
> >
> > Thanks for the clarifications. I'm going to keep my score as is. Please make sure you improve the presentation of your work following your response.

---

> > > ### Author Response · Authors · 2025-06-03
> > >
> > > Thanks for your comments. We’ve addressed all the concerns you raised and improved the presentation. We would appreciate it if you could update your score accordingly.

---

> > > ### Author Response · Authors · 2025-06-09
> > >
> > > Dear Reviewer,
> > >
> > > Thank you again for your valuable feedback. As the discussion period nearly comes to a close, we wanted to confirm that we have fully addressed all of your comments. If there are no remaining concerns, we would greatly appreciate your consideration in updating the score to reflect the improvements.
> > >
> > > Best regards,
> > >
> > > Authors

---

### Official Review · Reviewer_vYQH · 2025-05-09

**Rating:** 5
**Confidence:** 4
**Ethics Flag:** 1

**Summary:**

This paper studies Reinforcement Learning with Human Feedback (RLHF) for aligning LLMs with human preferences. They propose AUP-RLHF, which integrates user-level label DP into RLHF. They use outlier removal and adaptive sampling to ensure that the gradient is concentrated, and show an upper bound on the estimation error. Overall, this paper is clearly written and well motivated, but the experiments could be more thorough.

**Questions To Authors:**

Could the techniques be generalized to improve user-wise DP-SGD?

**Reasons To Accept:**

- User-level privacy is an important topic in LLM training.
- The authors present an improved algorithm for user-level label DP and give a theoretical upper bound on the estimation error.

**Reasons To Reject:**

Experiments could cover a wider range of hyperparameters.
- How do the results vary with the user contributions $m$?
- How much hyperparameter tuning was done? The results for DP training can be quite sensitive to the hyperparameters.

---

> ### Author Response · Authors · 2025-06-02
>
> - **Q1: Question about user-wise DP-SGD**
> >Could the techniques be generalized to improve user-wise DP-SGD?
>
> **Response to Q1**
>
> Thanks for the comment. Yes, our method can be used to user-wise DP-SGD with the same theoretical guarantee (under the same assumptions). However, as for different problems there may exist other methods. Thus, we cannot guarantee that our algorithm is better.
>
> ---
>
> - **W1: Question about user contributions $m$**
> > How do the results vary with the user contributions $m$?
>
> **Response to W1**
>
> According to Theorem 4, when the number of users $n$ is fixed, a larger $m$ results in a smaller upper bound on the estimation error, leading to better utility.
>
> To provide a more quantitative analysis, we present the following table showcasing the effect of varying $m$. We conduct experiments on IMDb Sentiment Generation using Gemma-2 2B, following the same settings as in Tables 3, 4, and 5.
>
> | m   | 10     | 100    | 250    | 500    |
> |-----|--------|--------|--------|--------|
> | reward           | 0.6402 | 0.8582 | 0.8539 | 1.2671 |
> | kl divergence    | 0.6960 | 0.9071 | 0.7434 | 0.8800 |
> | reward / kl divergence | 0.9198 | 0.9461 | 1.1486 | 1.4399 |
>
> As shown in the table, We observe that as the user contribution $m$ increases, the reward-to-KL divergence ratio improves at convergence, indicating that the model achieves better utility.
>
> ---
>
> - **W2: Question about hyperparameter tuning**
> >How much hyperparameter tuning was done? The results for DP training can be quite sensitive to the hyperparameters.
>
> From the DP perspective, we do not think this is a weakness based on the huge amount papers of DP-SGD.  Note that all DP training algorithms such as DP-SGD are sensitive to hyperparameters such as the clipping threshold. Private hyperparameter selection is an important project and it is still not well-understood.  Thus, almost all previous studies on DP-SGD only report the results base on the best  hyperparameter. In our paper, we follow the previous papers and our main experimental results all are based on on the best  hyperparameter, privately selecting the best  hyperparameter is beyond the scope of this paper.

---

> > ### Author Response · Authors · 2025-06-08
> >
> > Dear Reviewer vYQH,
> >
> > Thank you again for your valuable feedback. As the discussion period nearly comes to a close, we wanted to confirm that we have fully addressed all of your comments. If there are no remaining concerns, we would greatly appreciate your consideration in updating the score to reflect the improvements.
> >
> > We look forward to your feedback.
> >
> > Best regards,
> >
> > Authors

---

> > > ### Comment · Reviewer_vYQH · 2025-06-09
> > >
> > > Thank you for the clarifications. I will keep my score.

---

### Official Review · Reviewer_WBhh · 2025-05-13

**Rating:** 6
**Confidence:** 2
**Ethics Flag:** 1

**Summary:**

In this paper, the authors proposed AUP-RLHF, a user-level differentially private RLHF method. The authors first theoretically showed that existing DP methods that can achieve item-level differential privacy does not work well when naively extended to user level privacy, especially when each user make a lot of contributions. Then, the authors outlined AUP-RLHF in detail, and provided theoretical justification of the privacy and estimation error of the proposed algorithm. Finally, the author conducted experiments on RLHF on sentiment controlled generation and text summarization tasks, where the authors showed that AUP-RLHF outperforms other DP RLHF baselines under user-level DP settings, and that overall AUP-RLHF requires less noise addition compared to baselines.

**Questions To Authors:**

1. In all experiments and theoretical justifications, the authors assume that each user label the same amount of data; however, this is likely not going to be true in real world settings. If different users contribute a vastly different amount of data points to the dataset, how would that impact the performance and privacy guarantees of the proposed method?

2. Is it possible to provide a qualitative example of a model trained by AUP-RLHF on some user-contributed data that illustrates its privacy-preserving property? Currently all definitions about "user-level privacy" exists in terms of probability bounds, and it is kind of difficult for readers to get an intuitive grasp on the concept of user-level DP and why it matters.

3. How much does k matter in AUP-RLHF? According to line 256, the k for the experiments needs to be 3, but the k is set to 1 in all experiments. Also, strictly according to Algorithm 1, 1/2^k data would not be included in training (since each partition will have 1/2,1/4,...,1/2^k data), so in the experiments, when k=1, are you only using half the data to train?

**Reasons To Accept:**

1. There is extensive theoretical justification and proof for each claim (for both "user-level RR does not do well when m is large" and "AUP-RLHF is both user-level private and does not differ much from optimal model").

2. The experiment results clearly shows that AUP-RLHF outperforms other user-level DP algorithms.

**Reasons To Reject:**

1. The experiments are done on relatively simple tasks, and the authors did not provide qualitative examples of generated text from RLHF.

2. Presentation issues:

(1) Line 68: it is unclear what "outperforms well" means. Do you just mean "perform well"?

(2) Line 167: Is the feature bound L or B? They are being mixed up in this line.

---

> ### Author Response · Authors · 2025-06-02
>
> - **Q1: Question about different contributions**
> > In all experiments and theoretical justifications, the authors assume that each user label the same amount of data; however, this is likely not going to be true in real world settings. If different users contribute a vastly different amount of data points to the dataset, how would that impact the performance and privacy guarantees of the proposed method?
>
>   **Response to Q1:**
>   Under our theoretical framework, $m$ represents the minimum contributions per user. However, current approaches to user-level DP typically assume that each user contributes roughly the same amount of data [1][2].
>
>   [1] Levy et al., *Learning with User-Level Privacy*
>   [2] Asi et al., *User-level Differentially Private Stochastic Convex Optimization: Efficient Algorithms with Optimal Rates*
>
> ---
>
> - **Q2: Explanation about user-wise DP-SGD**
> > Is it possible to provide a qualitative example of a model trained by AUP-RLHF on some user-contributed data that illustrates its privacy-preserving property? Currently all definitions about "user-level privacy" exist in terms of probability bounds, and it is kind of difficult for readers to get an intuitive grasp on the concept of user-level DP and why it matters.
>
>   **Response to Q2:**
>   **On Qualitative Examples:**
>   The IMDb and TL;DR datasets follow the format described in Line 125, where each user is   $Z_i:=(s_{i,j},a^0_{i,j},a^1_{i,j},y_{i,j})_{j=1}^m $ .
>
>   Here, $s_{i,j}$ is the prompt, $a^0_{i,j}$, $a^1_{i,j}$ are candidate responses, and $y_{i,j} \in \\{0,1\\}$ indicates user preference.
>   Among these, only $y_{i,j}$ is considered a private attribute, while $s_{i,j}$, $a^0_{i,j}$, and $a^1_{i,j}$ are public. Thus, qualitative examples of RLHF outputs do not reveal the privacy guarantees of our method. Our guarantee ensures that model outputs are statistically similar regardless of whether a user's preference data is included in training.
>
>   **User-Level DP Intuition and Why It Matters:**
>   User-level DP guarantees that the presence or absence of an individual user's entire dataset does not significantly change the model's output, preventing adversaries from inferring whether a user contributed data. It is a stronger notion than record-level DP, which protects only single data points.
>
> ---
>
> - **Q3: Parameter $k$ in AUP-RLHF**
> > How much does $k$ matter in AUP-RLHF?
>
>   **Response to Q3:**
>   In the strongly convex case, $k$ arises from the localization framework [1], where the idea is to run Algorithm 2 for $k$ rounds, gradually refining the parameters to improve the theoretical bound.  However, $k$ is not a critical hyperparameter in practice. It is primarily introduced to facilitate theoretical analysis. As noted, terms like $\log \log (mn)$are $\mathcal{O}(1)$ and have negligible impact.  To simplify training, we set $k = 1$ and use the full dataset. Empirically, this already achieves performance close to the theoretical bound.
>
>   [1] Feldman et al., *Private Stochastic Convex Optimization: Optimal Rates in Linear Time*
>
> ---
>
> - **W1: About tasks and qualitative examples**
> > The experiments are done on relatively simple tasks, and the authors did not provide qualitative examples of generated text from RLHF.**
>
>   **Response to W1:**
>   The summarization task is not based on a small-scale dataset. The training set contains 92.9k preference pairs, and the validation set has 86.1k pairs, which is relatively large for alignment tasks.
>   Furthermore, summarization is not a trivial task. As shown in Figure 1 of *Learning to Summarize from Human Feedback*, the baseline model, even at a scale of 1B-7B, achieve only 0.6–0.7 win rates against human-written summaries.
>   In addition, the IMDb Sentiment Generation and TL;DR Summarization tasks are widely studied in previous RLHF literature and serve as commonly used benchmark tasks for evaluating RLHF [1] and DP-RLHF [2] methods.
>   The second part of the concern is addressed in Response to Q2.
>
>   [1] Rafailov et al., *Direct Preference Optimization: Your Language Model is Secretly a Reward Model*
>   [2] Wu et al., *Privately Aligning Language Models with Reinforcement Learning*
>
> ---
>
> - **W2: Presentation issues**
> > (1) Line 68: it is unclear what "outperforms well" means. Do you just mean "perform well"?
>   (2) Line 167: Is the feature bound L or B? They are being mixed up in this line.
>
>   **Response to W2:**
>   Thank you for the helpful comments.
>   (1) We agree and will revise “outperforms well” to “performs well” for clarity.
>   (2) Thank you for catching the typo. The correct feature bound is $L$.

---

> > ### Comment · Reviewer_WBhh · 2025-06-06
> >
> > Thank you for your response! My scores remain positive.

---

### Official Review · Reviewer_Bw9U · 2025-05-13

**Rating:** 7
**Confidence:** 2
**Ethics Flag:** 1

**Summary:**

This paper investigates user-level privacy guarantees for the user labels in RLHF. It presents a theoretical analysis of the upper bound and lower bound of the estimation error when using the random response (RR) mechanism to achieve user-level DP. The analysis reveals a large estimation error when each users' contribution becomes large, showing the RR-based method as infeasible for user-level privacy. It then presents AUP-RLHF, which is exhibits better utility than the RR algorithm in theoretical analysis and also outperforms User-wise DPSGD, RR, and Group Privacy in the experiments.

**Reasons To Accept:**

- Achieving user-level privacy guarantees for user labels in RLHF is an important problem. This paper makes novel contributions to this problem through both theoretical analysis and empirical improvements.
- The paper is well-written and easy to follow.
- The proposed method leverage outlier removal and adaptive sampling processes to achieve (ε, δ) user-level differential privacy guarantees with better utility.

**Reasons To Reject:**

N/A

---

> ### Author Response · Authors · 2025-06-02
>
> **Response**
>
> We would like to sincerely thank the reviewer for their positive feedback and thoughtful evaluation.
>
> We greatly appreciate the recognition of our contributions.

---

> > ### Comment · Reviewer_Bw9U · 2025-06-09
> >
> > I've read the response and other reviews and response and would like to remain with my positive score.

---

### Official Review · Reviewer_yUzC · 2025-05-22

**Rating:** 7
**Confidence:** 3
**Ethics Flag:** 1

**Summary:**

This paper applies differential privacy (DP) to reinforcement learning with human feedback (RLHF) from the perspective of protecting all of the labels for a user rather just individual users. This is more challenging that label-level (DP) as all of the data from a specific user provides more revealing information to a potentially snooper. The paper provides a theoretical deep-dive into the solution, which is further supported by experiments on two datasets.

**Questions To Authors:**

It would be useful to quantify the performance improvements at the end of the abstract.

Line 42: end the sentence at "user privacy". Adding "by introducing noise into the data" is just repeating the earlier part of the sentence.

Lines 64-66: What in particular is "sufficiently small"? I cannot tell if it is the SGD update steps, the noise add, or the concentration parameters.

Line 68: "which outperforms well" should be "which performs well".

Line 245: sigma is used to represent the noise, whereas earlier it was used to denote the sigmoid function in the Bradley-Terry model. Symbols should have a specific meaning in the paper to ensure there is no ambiguity.

Line 256: log appears twice when setting k, is this intentional?

Line 285: Will using GPT-4 as a proxy for humans mean the model will best to users that are most like GPT-4?

Line 296: Reorder the references to the figures so that they are in order (currently Figure 1, 4, 3).

Line 328: "multiple datasets". Technically two is multiple, but you could say "two commonly used datasets".

Line 331: Place a space before the word "Second".

**Reasons To Accept:**

Preference-learning is being increasing applied to align large models, including models already deployed to the general public for wide spread use. Guaranteeing the privacy of users via their feedback is important.

Many papers that focus on theoretical privacy guarantees for preference learning require overly simplifying assumption to be practically useful, such as the form of model that can be applied. Here the authors demonstrate application on reasonably sized language models.

**Reasons To Reject:**

Algorithm 2 -- Line 12:  a user is removed if they are considered an outlier. Does this mean that there are users for which the model cannot be aligned? Specifically, if the data for those users are effectively removed then the model cannot be adapted to them. Might this be problematic in the context of bias, where sub-populations of users might be harmed because their preferences are ignored?

---

> ### Author Response · Authors · 2025-06-02
>
> - **W1: Question about Algorithm 2**
> >Algorithm 2 -- Line 12: a user is removed if they are considered an outlier. Does this mean that there are users for which the model cannot be aligned? Specifically, if the data for those users are effectively removed then the model cannot be adapted to them. Might this be problematic in the context of bias, where sub-populations of users might be harmed because their preferences are ignored?
>
>  **Response to W1:**
>
> Thanks for the comment.
> (1) Even if a user is not selected in a particular mini-batch(line 4), they may still exhibit concentrated properties and be included in other batches.
>
> (2) User-level DP ensures that each user's contribution is upper bounded. As a result, if a user is an outlier, DP will downweight their influence. This inherent suppression of individual contributions is a key reason for the privacy-utility trade-off: compared to non-private settings, DP algorithms inevitably lead to reduced performance.
>
> Third, in terms of bias, since DP tends to suppress heavy-tailed or outlier contributions, it can increase the bias of the learning algorithm. This phenomenon has been extensively studied and documented in the literature such as [1][2].
>
> [1] Bagdasaryan, Eugene, Omid Poursaeed, and Vitaly Shmatikov. "Differential privacy has disparate impact on model accuracy." Advances in neural information processing systems 32 (2019).
>
>
> [2] Feldman, Vitaly. "Does learning require memorization? a short tale about a long tail." In Proceedings of the 52nd Annual ACM SIGACT Symposium on Theory of Computing, pp. 954-959. 2020.
>
> ---
> - **Q1: Question about Presentation**
> >Lines 64-66: What in particular is "sufficiently small"? I cannot tell if it is the SGD update steps, the noise add, or the concentration parameters.
>
>   >Line 256: log appears twice when setting k, is this intentional?
>
>   >Line 285: Will using GPT-4 as a proxy for humans mean the model will best to users that are most like GPT-4?
>
> **Response to Q1:**
>
> Thanks for the detailed suggestions. We will carefully revise the paper as you suggested.
>
> Lines 64–66: The term “sufficiently small” refers to the dded noise. We will clarify this in the revised version.
>
> Line 256 (setting $k$): The appearance of $\log \log(mn)$ is intentional, as it is required for our theoretical analysis.
>
> Line 285 (GPT-4 as human proxy): GPT-4 is only used as a proxy for evaluation purposes, not for generating preference training data, so the model will not be more aligned with users who are similar to GPT-4.

---

> > ### Comment · Reviewer_yUzC · 2025-06-02
> > **Thanks for the clarifying remarks**
> >
> > Thanks for taking the time to follow up with clarifications. Below are responses to these:
> >
> > > they may still exhibit concentrated properties and be included in other batches
> >
> > Thanks - I missed that this was batch-wise and that the user is excluded just from the batch. I had mistakenly understood that removing the user-data was globally.
> >
> > >  GPT-4 is only used as a proxy for evaluation purposes, not for generating preference training data, so the model will not be more aligned with users who are similar to GPT-4
> >
> > I see. But it may *appear* that for some users the model is better aligned because GPT-4 is used as a human proxy, even if the model is not technically better aligned to those users? I understand that using a model such as GPT-4 as a proxy for humans is a common approach, e.g., with AlpacaEval, just noting a possible limitation.
> >
> > I appreciate the clarification of my main concerns and will maintain my recommendation to accept the paper.

---

> > > ### Author Response · Authors · 2025-06-10
> > >
> > > Thank you for the thoughtful follow-up.
> > >
> > > We appreciate your engagement and support for accepting the paper.

---

### Official Review · Reviewer_c7jp · 2025-05-23

**Rating:** 4
**Confidence:** 3
**Ethics Flag:** 1

**Summary:**

This paper presents AUP-RLHF, a framework to ensure user-level label differential privacy in reinforcement learning from human feedback (RLHF), addressing a gap in current methods which mostly focus on item-level privacy. The authors provide a theoretical analysis, including a lower bound for user-level DP-RLHF and a provably tighter upper bound for their proposed method’s estimation error. The algorithm uses adaptive sampling and gradient concentration techniques to reduce noise while preserving privacy. Empirical results on IMDB sentiment generation and TL;DR summarization tasks demonstrate strong performance over existing baselines. Overall, the paper addresses an important and underexplored problem with a thoughtful approach and it is technically sound.

**Questions To Authors:**

1. Can authors clarify the computational overhead introduced by the adaptive concentration mechanism in AUP-RLHF compared to standard DP-SGD or RR-based approaches?
2. How sensitive is the performance of AUP-RLHF to the choice of hyperparameters?
3. Do the authors have empirical results on how AUP-RLHF performs in large-scale or more diverse RLHF tasks?

**Reasons To Accept:**

This paper tackles an important and timely problem in aligning large language models with human preferences while preserving user-level privacy—an area with growing practical relevance and limited prior work. The proposed AUP-RLHF framework is novel in its integration of adaptive sampling and gradient concentration into DP-SGD for RLHF, and it is supported by both theoretical guarantees and empirical improvements over existing user-level DP baselines.

**Reasons To Reject:**

While the paper addresses an important problem and offers a novel algorithm, several concerns limit its readiness for publication. First, the theoretical analysis, while rigorous, relies on strong assumptions, which may not hold in practice. Second, the empirical validation is limited in scope, using small-scale datasets and relatively simple tasks, making it unclear how well the method generalizes to more practical or large-scale RLHF settings. Third, the practical feasibility of the adaptive concentration mechanism is not thoroughly evaluated in terms of computational overhead or stability. Lastly, the overall utility gains over baselines are modest and may not justify the additional algorithmic complexity in real-world applications.

---

> ### Author Response · Authors · 2025-06-02
>
> - **W1: Question about assumptions**
> >First, the theoretical analysis, while rigorous, relies on strong assumptions, which may not hold in practice.
>
> **A:** Thanks for the comment. Note that all assumptions are commonly used in the previous literature on DP or RLHF. In detail, some user-level DP literature assumes that each user  holds $m$ i.i.d. samples from the same underlying distribution $P$ [1][2], while some works on RLHF and DP-RLHF assume boundedness and coverage of the feature space [3][4]. Thus, we cannot agree on your point.
>
> [1]Levy et al. *Learning with User-Level Privacy*
>
> [2]Asi et al. *User-level Differentially Private Stochastic Convex Optimization: Efficient Algorithms with Optimal Rates*
>
> [3]Zhu et al. *Principled Reinforcement Learning with Human Feedback from Pairwise or K-wise Comparisons*
>
> [4]Chowdhury et al. *Differentially Private Reward Estimation with Preference Feedback*
>
> ---
>
> - **W2: Question about experimental task**
> >Second, the empirical validation is limited in scope, using small-scale datasets and relatively simple tasks, making it unclear how well the method generalizes to more practical or large-scale RLHF settings.
>
> **Response to W2 and Q3:**
>
> Thanks for the comment.  The summarization task is not based on a small-scale dataset. The training set contains 92.9k preference pairs, and the validation set has 86.1k preference pairs, making it relatively large for alignment tasks. Additionally, the summarization task is not a simple task. As shown in Figure 1 of "Learning to summarize from human feedback," the baseline model, even at a scale of 1B-7B, achieves only a 0.6-0.7 win rate compared to human-annotated summaries. Moreover, the IMDb Sentiment Generation task and the TL;DR Summarization task are widely studied in previous RLHF literature and serve as commonly used benchmark tasks for evaluating RLHF[1] and DP-RLHF[2] methods.
>
> [1]Rafailov et al. *Direct Preference Optimization: Your Language Model is Secretly a Reward Model*
>
> [2]Wu et al. *Privately Aligning Language Models with Reinforcement Learning*
>
> ---
>
> - **W3: Question about computational overhead**
> >Third, the practical feasibility of the adaptive concentration mechanism is not thoroughly evaluated in terms of computational overhead or stability.
>
> **Response to W3 and Q1:**
>
> Thanks for the comment. .The main overhead of group privacy training methods is introduced by the per-example gradient clipping and noise addition. For the user-wise DP-SGD and AUP-RLHF, the main overhead is from the user-wise data sampling(line 4 in Algorithm 2), per-user gradient clipping(line 9 in Algorithm 4) or concentration scores computing(line 8 in Algorithm 2) and noise addition(line 15 in Algorithm 2)
>
> We report the average per-step training time (in seconds) for training the reward model using different methods, averaged over 100 steps 100 steps with a batch size of 500 on the IMDb Sentiment Generation task.
>
> Table: training time per step (seconds)
>
> | Random Response | Group Privacy | User-wise DPSGD | AUP  |
> |------------------|----------------|------------------|------|
> | 2.78             | 4.29           | 4.32             | 4.54 |
>
> From the table, we observe that AUP-RLHF incurs only a little computational overhead compared to Group Privacy and User-wise DP-SGD. The relatively short training time of RR is due to the absence of gradient clipping and noise addition.
>
> ---
>
> - **W4**
> >Lastly, the overall utility gains over baselines are modest and may not justify the additional algorithmic complexity in real-world applications.
>
> **A:** In IMDb Sentiment Generation task, as shown in Figure 1, AUP-RLHF consistently achieves higher rewards across all levels of KL divergence. Especially when KL divergence > 2, AUP-RLHF achieves up to 0.2 higher reward than other baselines.
>
> In the TL;DR summarization task , as shown in Figure 2, AUP-RLHF achieves a 63\% win rate in pairwise comparisons against the SFT model, representing a 7\% absolute improvement over User-wise DP-SGD , a 15\% improvement over Random Response, and a 16\% improvement over Group Privacy . These gains are not marginal.
>
> Moreover, AUP-RLHF introduces minimal computational overhead, with only a slight increase in running time.
>
> ---
>
> - **Q2**
> >How sensitive is the performance of AUP-RLHF to the choice of hyperparameters?
>
> **A:** From the DP perspective, we do not think this is a weakness based on the huge amount papers of DP-SGD.  Note that all DP training algorithms such as DP-SGD are sensitive to hyperparameters such as the clipping threshold. Private hyperparameter selection is an important project and it is still not well-understood.  Thus, almost all previous studies on DP-SGD only report the results base on the best  hyperparameter. In our paper, we follow the previous papers and our main experimental results all are based on on the best  hyperparameter, privately selecting the best  hyperparameter is beyond the scope of this paper.

---

> > ### Comment · Reviewer_c7jp · 2025-06-06
> >
> > Thanks to the authors for the explanation. However, these explanations do not address the majority of my concerns. Therefore I will maintain the original score.

---

> > > ### Author Response · Authors · 2025-06-06
> > >
> > > Thanks for the comment. In our rebuttal, we have carefully addressed all the concerns raised in your original review, point by point. You mentioned that “these explanations do not address the majority of my concerns.” Could you kindly clarify which specific concerns you feel remain unresolved?

---

> > > ### Author Response · Authors · 2025-06-09
> > >
> > > Dear Reviewer,
> > >
> > > Thank you again for your valuable feedback. As the discussion period nearly comes to a close, we wanted to confirm that we have fully addressed all of your comments. If there are no remaining concerns, we would greatly appreciate your consideration in updating the score to reflect the improvements.
> > >
> > > Best regards,
> > >
> > > Authors

---

### Official Review · Reviewer_1im1 · 2025-05-25

**Rating:** 7
**Confidence:** 3
**Ethics Flag:** 2

**Summary:**

- Quality and clarity: this paper proposes AUP-RLHF algorithm that integrates user-level label differential privacy into RLHF. It contains theoretical analysis for upper and lower bounds; as well as empirical experiment on PPO for sentiment and summarization tasks. The content is rich and clear.

- Originality: It applies the existing user-level private mean estimation on DP-SGD for the new task of user preference label privacy protection and derives bounds for preference label DP.

- Significance: this paper focuses on protecting user preference label on pair-wise preference data. It is a first attempt to protect this type of preference privacy.

**Questions To Authors:**

Could you provide more explanations for how you conduct experiments for user preference label protection, for example how well is the preference label protected (what is the baseline) (and the case without DP)? What does timestep mean in Figure 4?

**Reasons To Accept:**

- Clear writing with both theoretical analysis for bounds and empirical results.
- New task of user preference label privacy protection.

**Reasons To Reject:**

- Based on existing user-level DP methods; some small modifications to adapt to new objective (algorithm-wise).

---

> ### Author Response · Authors · 2025-06-02
>
> - **W1: Question about algorithm**
> >Based on existing user-level DP methods; some small modifications to adapt to new objective (algorithm-wise).
>
> **Response to W1:**
>
> Thanks for the comment. Our contribution lies in proposing a theoretical framework to address the problem of user-level privacy in the RLHF setting. The AUP-RLHF algorithm we introduce is specifically designed for RLHF. Developing a better user-level DP-SGD algorithm is not the focus of our contribution.
>
> ---
>
> - **Q1: Question about label protection**
> >Could you provide more explanations for how you conduct experiments for user preference label protection, for example how well is the preference label protected (what is the baseline) (and the case without DP)? What does timestep mean in Figure 4?
>
> **Response to Q1:**
>
> (1)We include Group Privacy, User-wise DP-SGD, and Randomized Response (RR) as baselines. RR is described in detail in Section 4. Group Privacy leverages the composability property of DP, where a record-level DP mechanism can be directly extended to provide user-level privacy guarantees.
>
> User-wise DP-SGD modifies the original DP-SGD by incorporating user-level sampling and per-user gradient clipping. However, unlike our proposed AUP-RLHF, it lacks adaptive sampling based on the concentration score, which limits its ability to achieve a better privacy-utility trade-off.
>
> For the IMDb sentiment generation task, we use the ground truth reward model to compute the reward of the model-generated responses.
> For the TL;DR summarization task, we compare the responses from RLHF-tuned LLMs and SFT models to see which are more preferred by human annotators, and calculate the win rate.
> Both results demonstrate that our AUP-RLHF outperforms the other baselines.(Figure 1 and 2)
>
> (2)Our preference label protection is rigorously guaranteed by differential privacy (DP) theory, as formalized in Theorem 4. Moreover, prior empirical studies have shown that label-level DP can effectively defend against membership inference attacks (MIA) compared with the case without DP[1], further demonstrating our method's ability to protect preference labels.
>
> (3)In Figure 4, "timestep" refers to the number of training steps in the PPO training phase of RLHF.
>
> [1]Wu et al. Does Label Differential Privacy Prevent Label Inference Attacks?

---

> ### Author Response · Authors · 2025-06-10
>
> Dear Reviewer,
>
> Thank you again for your valuable feedback. As the discussion period nearly comes to a close, we wanted to confirm that we have fully addressed all of your comments. If there are no remaining concerns, we would greatly appreciate your consideration in updating the score to reflect the improvements.
>
> Best regards,
>
> Authors

---

> > ### Comment · Reviewer_1im1 · 2025-06-10
> >
> > Thank you for your explanations and I would maintain the score. I understand that your paper focuses on the task of user-level privacy in the RLHF setting so your developments are more specific. I would recommend to make the writing about comparison / experimental settings clearer.

---

### Decision · Program_Chairs · 2025-07-08

**Decision:**

Accept

**Comment:**

This paper presents a framework for addressing user-level label differential privacy in RLHF. The work is well-grounded with a solid theoretical analysis, establishing both lower and upper bounds for estimation error, and is supported by empirical results on sentiment generation and summarization tasks that demonstrate a superior privacy-utility trade-off compared to several baselines. While some reviewers initially raised valid concerns regarding experimental scope and practical assumptions, these concerns are addressed by additional results and detailed clarifications. This paper makes important contribution to the field of trustworthy AI, outweigh the remaining minor concerns.